# Enhanced FIB-SEM systems for large-volume 3D imaging

C Shan Xu[1]*, Kenneth J Hayworth[1], Zhiyuan Lu[1,2], Patricia Grob[3], Ahmed M Hassan[3†], José G García-Cerdán[4], Krishna K Niyogi[4,5], Eva Nogales[3,5], Richard J Weinberg[6], Harald F Hess[1]

[1]Janelia Research Campus, Howard Hughes Medical Institute, Ashburn, United States; [2]Department of Psychology and Neuroscience, Dalhousie University, Halifax, Canada; [3]Howard Hughes Medical Institute, Molecular and Cell Biology Department, University of California, Berkeley, United States; [4]Howard Hughes Medical Institute, Plant and Microbial Biology Department, University of California, Berkeley, United States; [5]Lawrence Berkeley National Laboratory, Berkeley, United States; [6]Department of Cell Biology and Physiology, University of North Carolina, North Carolina, United States

**Abstract** Focused Ion Beam Scanning Electron Microscopy (FIB-SEM) can automatically generate 3D images with superior z-axis resolution, yielding data that needs minimal image registration and related post-processing. Obstacles blocking wider adoption of FIB-SEM include slow imaging speed and lack of long-term system stability, which caps the maximum possible acquisition volume. Here, we present techniques that accelerate image acquisition while greatly improving FIB-SEM reliability, allowing the system to operate for months and generating continuously imaged volumes > $10^6$ $\mu m^3$. These volumes are large enough for connectomics, where the excellent z resolution can help in tracing of small neuronal processes and accelerate the tedious and time-consuming human proofreading effort. Even higher resolution can be achieved on smaller volumes. We present example data sets from mammalian neural tissue, *Drosophila* brain, and *Chlamydomonas reinhardtii* to illustrate the power of this novel high-resolution technique to address questions in both connectomics and cell biology.

*For correspondence: xuc@ janelia.hhmi.org

**Present address:** †Department of Biomedical Engineering, University of Texas at Austin, Austin, United States

**Competing interests:** The authors declare that no competing interests exist.

## Introduction

Many modalities of electron microscopy (EM) can probe cellular structure at the nanometer scale. However, despite considerable progress over the past decade in developing high-resolution 3D imaging, there remain important limitations, reflecting an inherent trade-off between resolution and the size of the 3D volume. Different currently available EM methods, and their advantages and disadvantages have been reviewed recently (*Briggman and Bock, 2012*; *Titze and Genoud, 2016*). For demanding applications such as tracing neuronal processes in three dimensions, high resolution in the z axis, in addition to the xy plane, is critical (*Lichtman and Denk, 2011*; *Meinertzhagen, 2016*). FIB-SEM offers exactly this capability, with x, y, and z resolution all <10 nm (*Knott et al., 2008*; *Xu and Hess, 2011*), in addition to other significant advantages, such as superior registration and fully automated operation. However, to date the FIB-SEM approach has seldom been used in neuroscience because of its severe volume limitation (*Briggman and Bock, 2012*; *Knott et al., 2008*; *Helmstaedter, 2013*; *Denk et al., 2012*), typically less than the extent of a single neuron.

FIB-SEM was originally developed for semiconductor and material research applications without requirements for imaging large volumes; only in the past decade has it been explored as a tool for 3D biological imaging (*Knott et al., 2008*; *Xu and Hess, 2011*; *Narayan and Subramaniam, 2015*;

**eLife digest** Precise three-dimensional imaging can help make sense of microscopic details in biology. These images are usually built up from many two-dimensional images stacked on top of each other. One approach for examining particularly fine details, such as the connections between nerve cells in the brain, is called focused ion beam scanning electron microscopy (or FIB-SEM for short). This approach works by creating an image of the surface layer of a sample, which is then stripped away using a beam of charged particles to reveal the layer beneath. The new surface can then be imaged and so on, through the whole sample.

Unfortunately, FIB-SEM devices are currently slow and can only run for a short time, leading to a lack of continuity in the stack of images. FIB-SEM would allow faster, more accurate and detailed studies of connections between brain cells, and other elaborate biological systems, if the technology could be made faster and more reliable over months of continuous operation. The current technical challenge is to create a system that can, for example, successfully image and analyse all the connections between the more than 100 thousand cells that make up the brain of a fruit fly – a common model organism in neurobiology.

Xu et al. aimed to create a technique to image a complete fly brain, with gaps of just 8 nanometres between each image in a stack, within a reasonable timeframe. By improving how FIB-SEM signals are detected, making use of advances in ion beam controls, and by engineering ways to recover from system malfunctions, Xu et al. developed an enhanced FIB-SEM device. To demonstrate its value, the new technology was used to create images of a third of a fruit fly's brain, parts of a mouse's brain, and cells of a single-celled alga called *Chlamydomonas reinhardtii*.

The results show that large and complex samples can be successfully imaged in their entirety to adequate detail, enabling high-quality reconstruction of the connections between nerve cells. The level of detail, which can be further increased for smaller samples, offers advantages in precision and image quality over other comparable techniques. As well as helping to study the brain, this approach could also be used to examine details inside cells. Future work to advance this technology will enable larger and more complete imaging of elaborate biological structures.

*Wei et al., 2012*). The typical 3D FIB-SEM procedure uses a focused ion beam to ablate a few nanometer layer from the specimen block-face, followed by SEM imaging of the freshly exposed surface. These steps cycle continually until the entire 3D volume is ablated and imaged. The distinctive advantage of FIB-SEM is the fine z thickness removed with each step, which gives z resolution down to a few nanometers. In contrast, the current state-of-the-art technologies based on diamond knife sectioning (*Hayworth et al., 2006*) or diamond knife block-face removal (*Denk and Horstmann, 2004*; *Wanner et al., 2015*) lose consistency when attempting z steps between adjacent images below 20 nm. Deconvolution based on multiple images with varying landing energies can improve z resolution on thin sections (*Boughorbel et al., 2012*; FEI Teneo VS Technology: http://www.fei.com/teneo-for-life-sciences/), but only to a limited extent. Electron tomography, based on tilting thin sections in TEM, provides excellent z resolution but becomes impractical for reconstructing thick samples due to the tedious stitching requirements of a long series of these tomograms from sequential sections (*Soto et al., 1994*).

Most of these EM techniques yield very different resolutions in the x, y, and z directions, and reduced resolution in any one axis can introduce

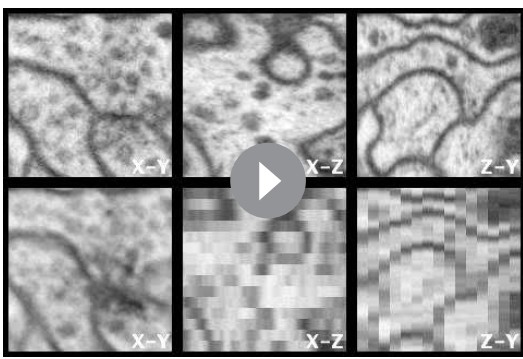

**Video 1.** Three-dimensional x,y,z data showing 4 nm voxels over 600 nm range of *Drosophila* neuropil with isotropic resolution (top row), and a section where the data is binned together in z to form 4 x 4 x 40 nm³ voxels, to emulate standard TEM sections.

significant burden to subsequent image processing and analysis such as segmentation and proof-reading in connectomics studies. Figure 2 and *Video 1* illustrate this shortcoming by comparing experimental isotropic 3D data and simulated 3D data with a 40 nm sampling interval in z. The anisotropic image stack was generated by averaging 10 frames of the nearly isotropic 4 nm voxel data set along the z-axis to emulate data from a 40 nm section. This emulation is not identical to a TEM image stack collected from 40-nm-thick sections due to different contrast mechanisms provided by backscattered and transmitted electrons. Nevertheless, it offers a first-order comparison between the two imaging modalities, showing the limitations of TEM's anisotropic data, where details become poorly resolved in re-sliced non-imaging planes. For example, the xz and zy planes show much degraded resolution compared to the xy image planes, due to poor z resolution. This is particularly troublesome in connectomics (the study of neural connectivity), which needs to resolve fine neural processes parallel to the xy imaging plane. In an isotropic data set, there will be no degradation of resolution at any re-sliced planes at random angles. This feature has substantial benefits for tracing neuronal circuitry, where one needs to follow fine processes that are orientated at random angles. Accordingly, we use a unifying resolution metric, where resolution is defined by the worst case in the x, y, or z direction. To elucidate biological structure, it is helpful to render 3D images, without any axial bias, with isotropic resolution. The worst-case axial resolution then dictates the appropriate minimal isotropic voxel size for sampling and rendering.

The graphical summary (*Figure 1*), which shows the operating regimes of the different EM methods in terms of sample volume and minimum isotropic resolution, identifies an important region of resolution-volume space that remains inaccessible with current techniques. FIB-SEM provides a logical probe for this region, but until now, technical obstacles have blocked its use. The most prominent such obstacle is the volume limitation, dictated by the limited imaging speed and the limited duration of smooth and consistent ablation. Because the process is destructive, there is little room for error in the ablation-imaging cycle, which requires virtually perfect continuity and consistency. Here, we describe a series of measures that address these limitations, thus transforming FIB-SEM into a tool capable of probing this 'dark' region of resolution-volume space. We also provide examples to illustrate the potential of large volume FIB-SEM for both neurobiology and cell biology.

## Results and discussion

### Technological improvements

With connectomics in mind, we designed a customized FIB-SEM system to address the prevailing deficiencies in imaging speed and duration. Our new system incorporates many prior improvements in ion beam and electron microscopy, and new advances that represent key enabling features for large-volume 3D imaging. The two most important technological advances are imaging speed improvement and error detection followed by seamless recovery. Negative sample biasing, an established procedure in scanning electron microscopy, is typically used to improve resolution (*Bouwer et al., 2016*). However, we have found that a moderate positive bias provides a simple and straightforward way to filter out secondary electrons. This scheme transforms a traditional in-column (InLens) detector into an effective backscattered electron detector that captures a larger fraction of the backscattered electrons, resulting a ~ 10x improvement of imaging speed without contrast degradation compared to a traditional energy-selective backscattered (EsB) detector alone (see Figure 11j).

Considerable engineering effort is required to gain major improvements in system reliability. Most of the individual components, especially those achieved via software controls, are not themselves innovative, but their combined effect is transformative. First, multiple layers of error and disturbance protection, including refinements and additions in hardware, software, and utilities, were introduced to prevent catastrophic failures such as an uncontrolled sample ablation. Detailed descriptions of these strategies can be found in the Technology and Methods section. Second, an extension of the closed-loop control of the ion beam (first introduced by Denk and co-workers [*Boergens and Denk, 2013*]) maintained stability, and enabled a seamless restart of the imaging cycle after interruptions. Third, the FIB column was repositioned to be 90 degrees from the SEM column instead of the standard 52–55 degrees; this enabled a shorter working distance, which enhanced the imaging quality. Together these modifications provide a speedy system with overall

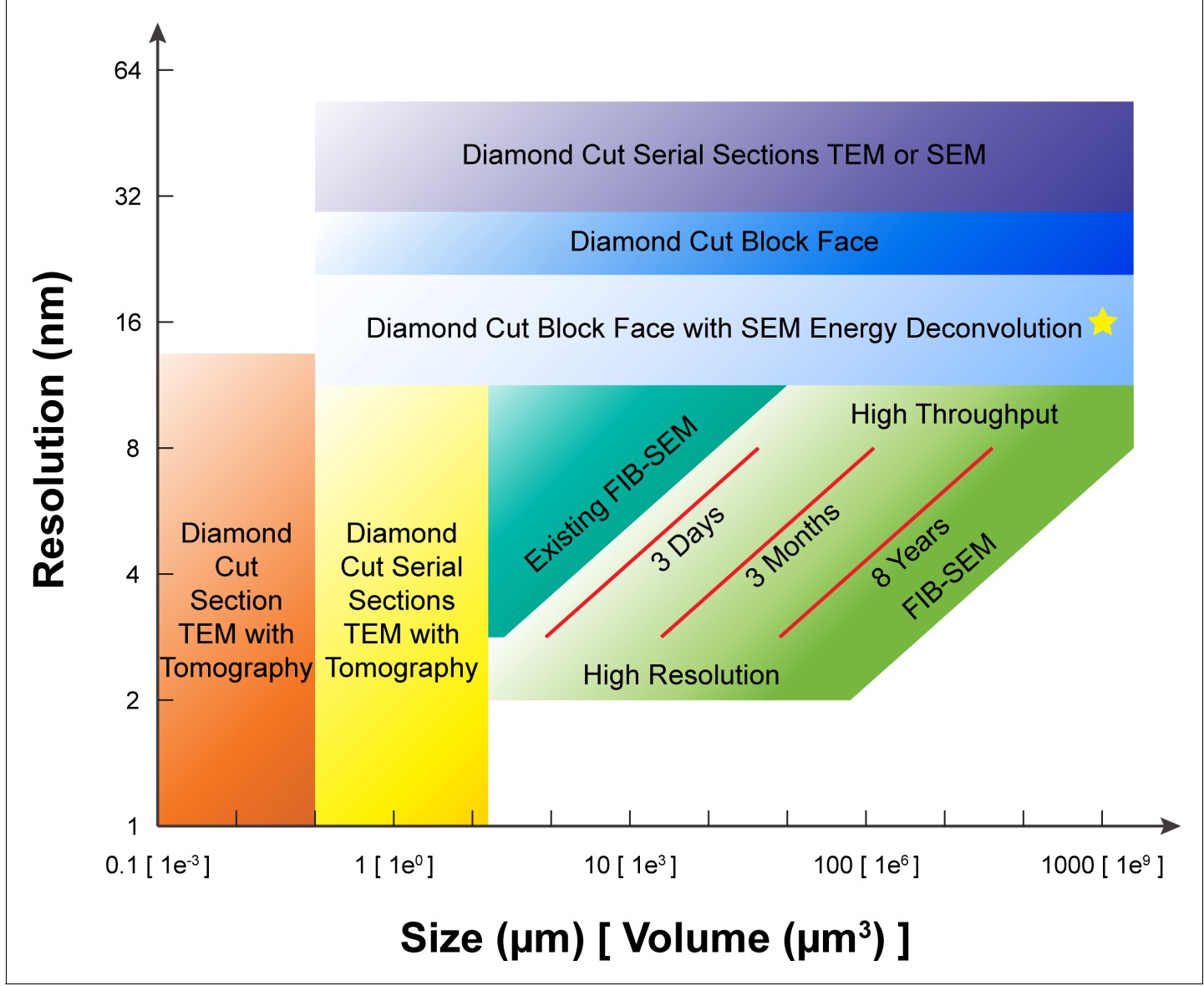

**Figure 1.** A comparison of various 3D imaging technologies in the application space defined by resolution and total volume. The resolution value indicated by the bottom boundary for each technology regime represents the minimal isotropic voxel it can achieve, while the size value indicated by the right boundary is the corresponding limit in total volume. An expansion in total volume and improvement in resolution of FIB-SEM would fulfill a desired space at the lower right corner, not yet accessible with any existing technology. The three red diagonal constant imaging time contours indicate the general trade-off between resolution and total volume during FIB-SEM operations of 3 days, 3 months, and 8 years, respectively, using a single FIB-SEM system. These contours are sensitive to staining quality and contrast. The yellow star indicates the intercept between the extrapolated 8-year contour and 1 mm$^3$ volume. Considering the hot-knife overhead and machine maintenance downtime, a more realistic estimate would be ~3 years using 4 FIB-SEM systems. The boundaries of the different imaging technologies outline the regimes where they have a preferential advantage, though in practice there is considerable overlap and only a fuzzy boundary.

virtual reliability exceeding that of its individual components. Biological volumes as large as 1 million µm$^3$ containing biologically meaningful neuronal elements, such as individual modules in a *Drosophila* brain, can now be routinely acquired in a few months. Furthermore, with a previously reported ultrathick partitioning technique, even larger volumes (e.g. an entire *Drosophila* brain) could potentially be subdivided into small pieces, then imaged with multiple FIB-SEM systems running in parallel (*Hayworth et al., 2015*).

The capacity for sustained operation also opens an opportunity for cell biologists to explore research topics dependent upon the ability to resolve fine (<5 nm) features in 3D. In such applications, a slower high-resolution imaging modality can acquire volumes up to tens of micrometers. By providing straightforward generation of large high-resolution isotropic FIB-SEM datasets, the strategies outlined here can provide clear visualization of complex fine-grained biological structures, permitting exploration of novel elements of cellular architecture.

## FIB-SEM for high-throughput connectomics

To assess the value of FIB-SEM's superior z-axis resolution for connectomics research, a portion of a *Drosophila* optic lobe (*Takemura et al., 2015*) containing seven medulla columns was imaged at an isotropic resolution of $10 \times 10 \times 10$ nm$^3$ voxels. The entire volume of $30 \times 30 \times 60$ µm$^3$ was acquired over 2 weeks, and then segmented, annotated, and proofread. We compared these data with those from a previous study of equivalent material studied with classical serial-section TEM performed on 40 nm-thick sections (*Takemura et al., 2013*). While TEM of sections can take beautiful images of dendrites and synapses, many important details will be obscured if they are oriented in the wrong direction, for example a fine dendrite with its axis running parallel to the image plane of a section is easily lost (*Figure 2*). Accordingly, 50% more synaptic connections were detected within a single medulla column in the FIB-SEM data set than in the TEM image stack (*Takemura et al., 2015*). These improvements in accuracy provided by FIB-SEM data analysis represent a gold standard, useful for understanding the level of completeness of a connectome derived from TEM sections. Furthermore, by imaging the intact block-face, registration and alignment is easy, unlike in serial section TEM where section tears, scratches, and distortions require complex corrections. Finally, the rate of volume reconstruction, which includes synapse identification, segmentation, and

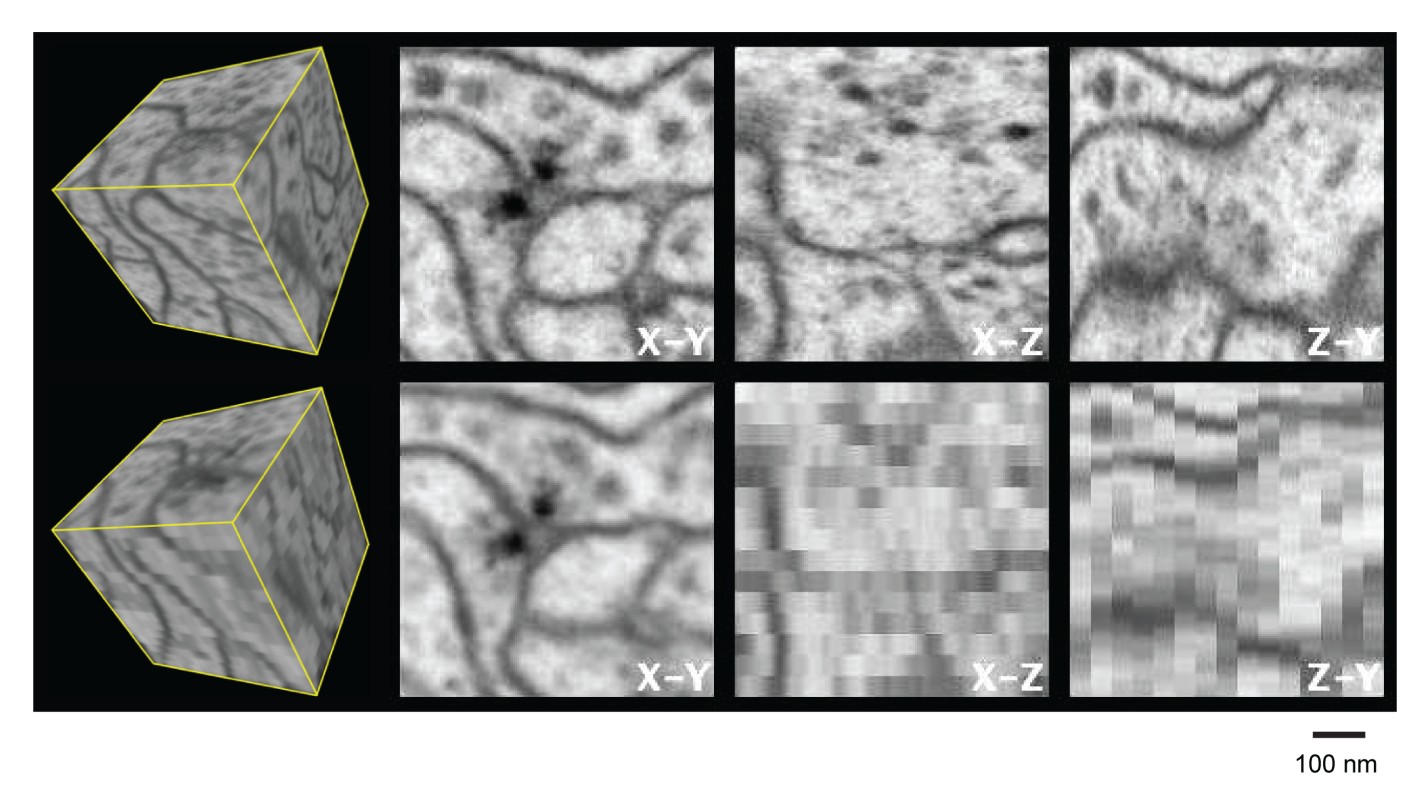

100 nm

**Figure 2.** Three orthogonal views of a (600 nm)$^3$ block of *Drosophila* neuropil with isotropic 4 nm voxels and anisotropic ($4 \times 4 \times 40$ nm$^3$) voxels derived from the isotropic data to emulate 40-nm section data. *Video 1* corresponds to this Figure. Scale bar, 100 nm.

proofreading was ~3–5x faster for the FIB-SEM data (*Plaza, 2014*), thanks to finer z resolution and better image registration.

The same FIB-SEM system has been used to image other parts of the *Drosophila* brain of comparable size, including the antenna lobe and mushroom body (Takemura et al., submitted), with comparable improvements in both speed and accuracy. To evaluate the suitability of the customized FIB-SEM system for imaging even larger volumes, the *Drosophila* optic lobe, including medulla, lobula, and lobula plate, was imaged at $8 \times 8 \times 8$ nm$^3$ voxel resolution over a 100 day period, yielding a final four terabyte image volume of about $180 \times 100 \times 50$ µm$^3$ (50 µm in the direction of the FIB beam), shown in *Figure 3*. We encountered multiple unplanned system failures ranging from replacement of the SEM field emitter tip to complete pump failure, in addition to more than 20 planned interruptions reflecting the need to replenish the FIB source every 4–6 days. With a standard FIB-SEM system, these interruptions would have led to multiple gaps and other defects in the final image stack that would make it virtually impossible to perform any large-scale reconstructions of connectivity. However, our customized FIB-SEM system was designed to pause the system promptly when interruptions occurred, and to resume seamlessly after the system returned to normal operation. Only one noticeable imperfection associated with system shutdown and source replacement was passed down to the final 3D volume images. Data from this run are shown in *Figure 3b,c*, which render x-z and y-z re-sliced views of a large volume that includes medulla, lobula, and lobula plate. Because the voxels were isotropic, re-slicing did not degrade image resolution. Plasma membranes, presynaptic T-bars, and postsynaptic densities were clearly visible in both renderings. Importantly, there were no visible discontinuities due to system interrupts other than the one noted above (*Video 2*). The robust handling of both scheduled and random interruptions enables long-term operations spanning months of imaging.

This image volume can be segmented and proofread in its entirety to create dense reconstructed data sets where > 90% of the synaptic contacts are identified and assigned to reconstructed neurons. *Figure 4* illustrates this for two compartments of the fly brain: the medulla and mushroom

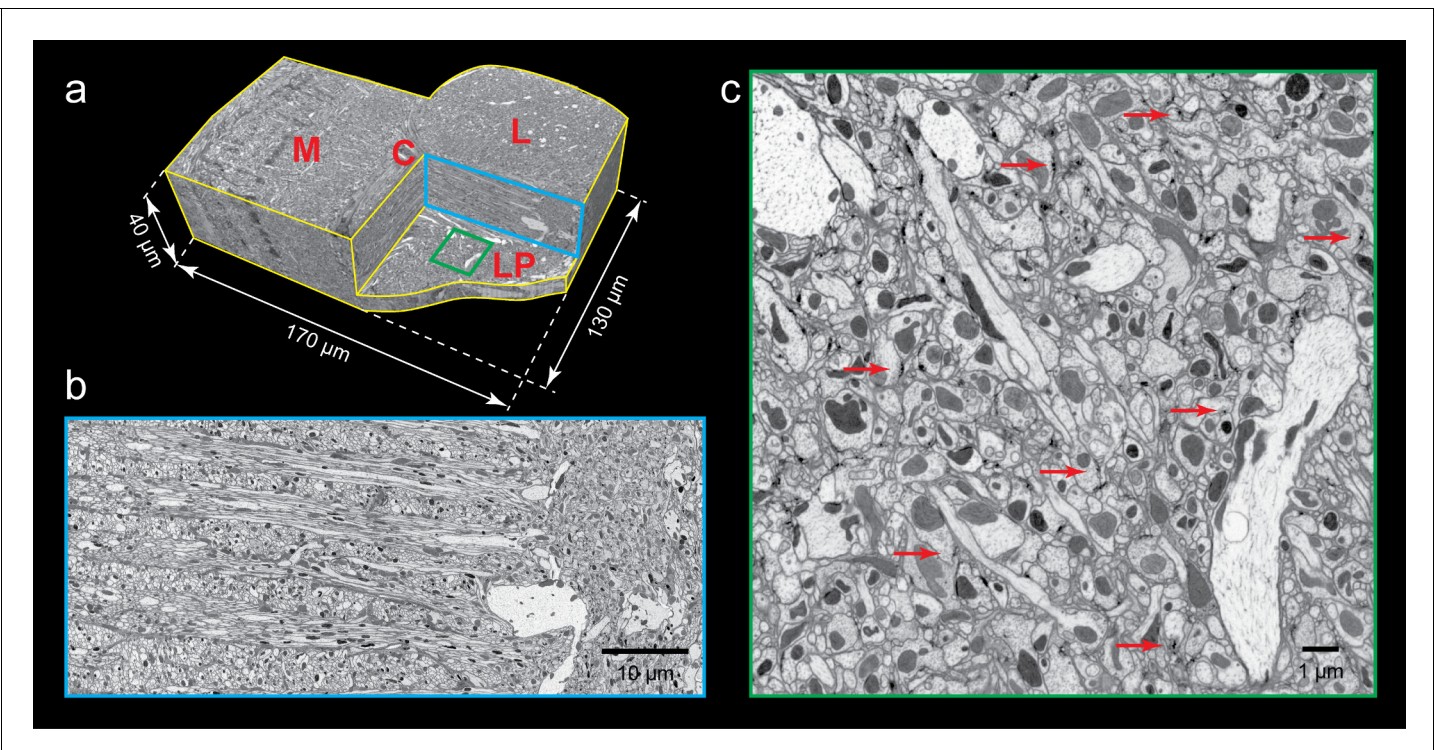

**Figure 3.** FIB-SEM images of *Drosophila* optic lobe. (**a**) A cropped FIB-SEM volume showing medulla (M), lobula (L), lobula plate (LP), and chiasm (C). (**b**) An enlargement of the blue cross-section in (**a**) showing a re-sliced y-z plane. Scale bar, 10 µm. (**c**) An enlargement of the green box in (**a**) showing a re-sliced x-z plane where very fine neural processes are visible. Red arrows indicate synaptic structures. Scale bar, 1 µm.

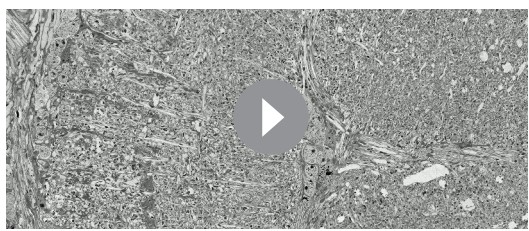

**Video 2.** Re-sliced view of *Drosophila* optic lobe showing medulla (left), lobula (upper right), lobular plate (lower right). 150 × 64 × 40 µm³ region with 10x zoom, 8 × 8 × 8 nm³ voxel.

body. In the cross-sections of *Figure 4b,c*, segmented regions assigned to specific neurons that are contained in the imaged volume are colorized in green, while processes traced to neurites that extend tangentially outside the imaged volume are colorized in yellow. Only a small fraction of the area remains red, corresponding to 'orphaned' incompletely reconstructed neuron fragments. Note that in the rendering of *Figure 4c* all cells in the mushroom body have been identified. The few examples in the literature that have achieved comparable levels of volume completeness all required multi-year proofreading or tracing efforts by large teams (*Meinertzhagen, 2016*). These dramatic improvements in both the overall efficiency of the reconstruction effort and the degree of completeness of the connectome are important benefits of isotropic block-face FIB-SEM data.

For even larger samples with the typical electron dose of our standard SEM imaging condition, when dimensions in the direction of the FIB beam exceed 60–100 µm a FIB milling instability emerges, producing curtains and waves of non-uniform material removal that limit data quality (*Lemmens et al., 2011*). We have addressed this by a hot knife partitioning method

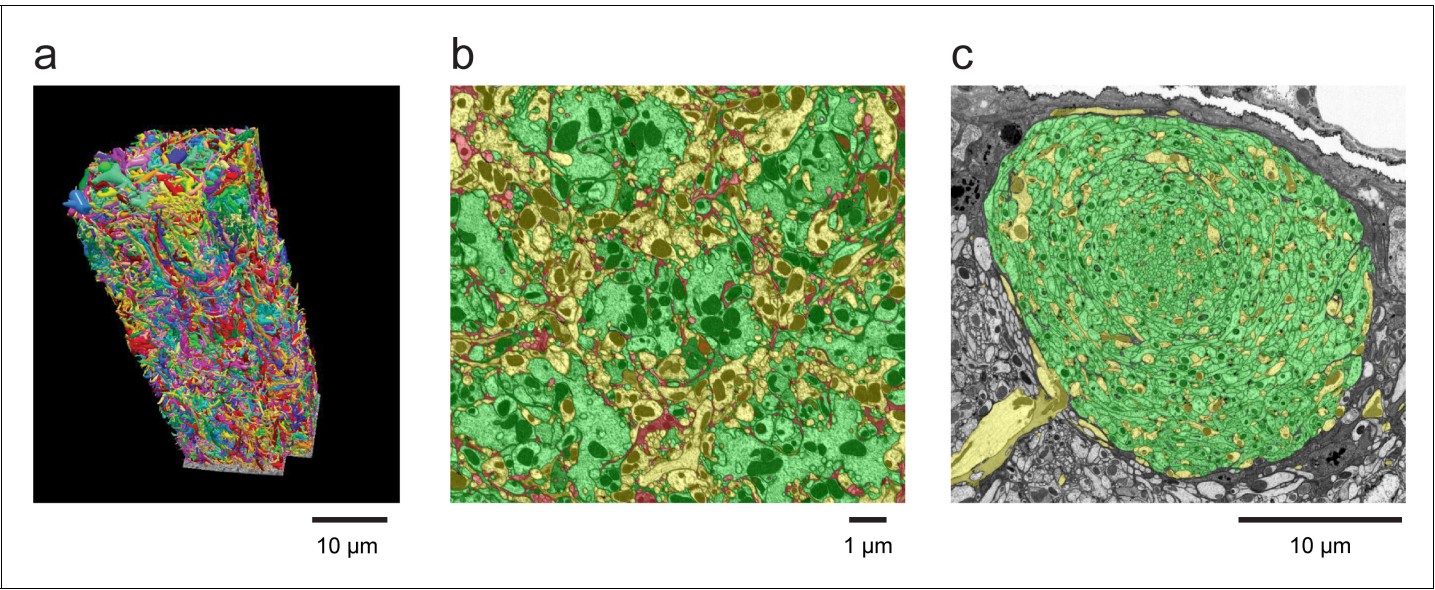

a

b

c

10 µm          1 µm          10 µm

**Figure 4.** Examples of densely reconstructed data. (a) A 3D rendering of seven columns of the medulla of the *Drosophila* optic lobe from FIB-SEM showing reconstructed neurons from a ~ 30,000 µm³ volume. This reconstruction, which required over five man-years of effort, was ~3–5x faster than a comparable optic lobe reconstruction using an image stack from serial-section TEM, for which we have reconstructed a single medulla column (*Takemura et al., 2013*). Scale bar, 10 µm. (b) A cross-section of the neuropil of the medulla in the optic lobe of *Drosophila,* showing the high degree of reconstruction completeness that is possible with FIB-SEM data. The hexagonal periodicity reflects the hexagonal pattern of the ommatidia of the fly's retina. The colors illustrate how all neural processes have been assigned. Green indicates various identified columnar input neurons contained within this volume, and yellow indicates axons and arbors of various medulla neurons that branch into or out of this volume. The small remainder (shown in red) highlights the 'left over' parts, including unidentified and orphaned fragments of neurons and glial processes. Well over 90% of the neuropil volume could be reconstructed and assigned to specific neurons. Scale bar, 1 µm. (c) Cross-section of the neuropil of the mushroom body of *Drosophila*. Notice that virtually all processes in this section have been identified and colorized green (to denote Kenyon cells) or yellow (for other identified mushroom body neurons). The only 'left over' uncoded processes are a few thin fragments dispersed within the mushroom body boundary that could not be confidently assigned to a specific cell. The mushroom body volume was comparable to the seven-column medulla volume and required a comparable reconstruction effort. Scale bar, 10 µm. Image process, segmentation, and 3D rendering provided by the Janelia FLYEM team, see Acknowledgements.

(*Hayworth et al., 2015*) that subdivides a larger sample into 20 µm-thick slabs. The method preserves sample quality to within ~20 nm of the cut surface, enabling effective stitching of 3D connectomic data across the cut slabs. An overview of hot knife partitioning and imaging results are shown in *Figure 5*. We have imaged nine slabs of 20 × 250 × 250 µm³ with two FIB-SEM systems in parallel, representing a total volume of over 10 × 10⁶ µm³ that spans the key central complex components of the fly brain. One of these slabs is shown in *Video 3*. A complete fly brain of 30 slabs could be imaged in ~5 FIB-SEM-years of acquisition time; because the approach is scalable, multiple FIB-SEM machines could be used to reduce the total time required. Though substantial, this acquisition duration is dwarfed by the overriding component of the dense connectome pipeline: tracing and proofreading of the segmented data, which can take two to three orders of magnitude more man-years (*Plaza, 2014*)! Efficient proofreading thus requires teams of hundreds to thousands of people

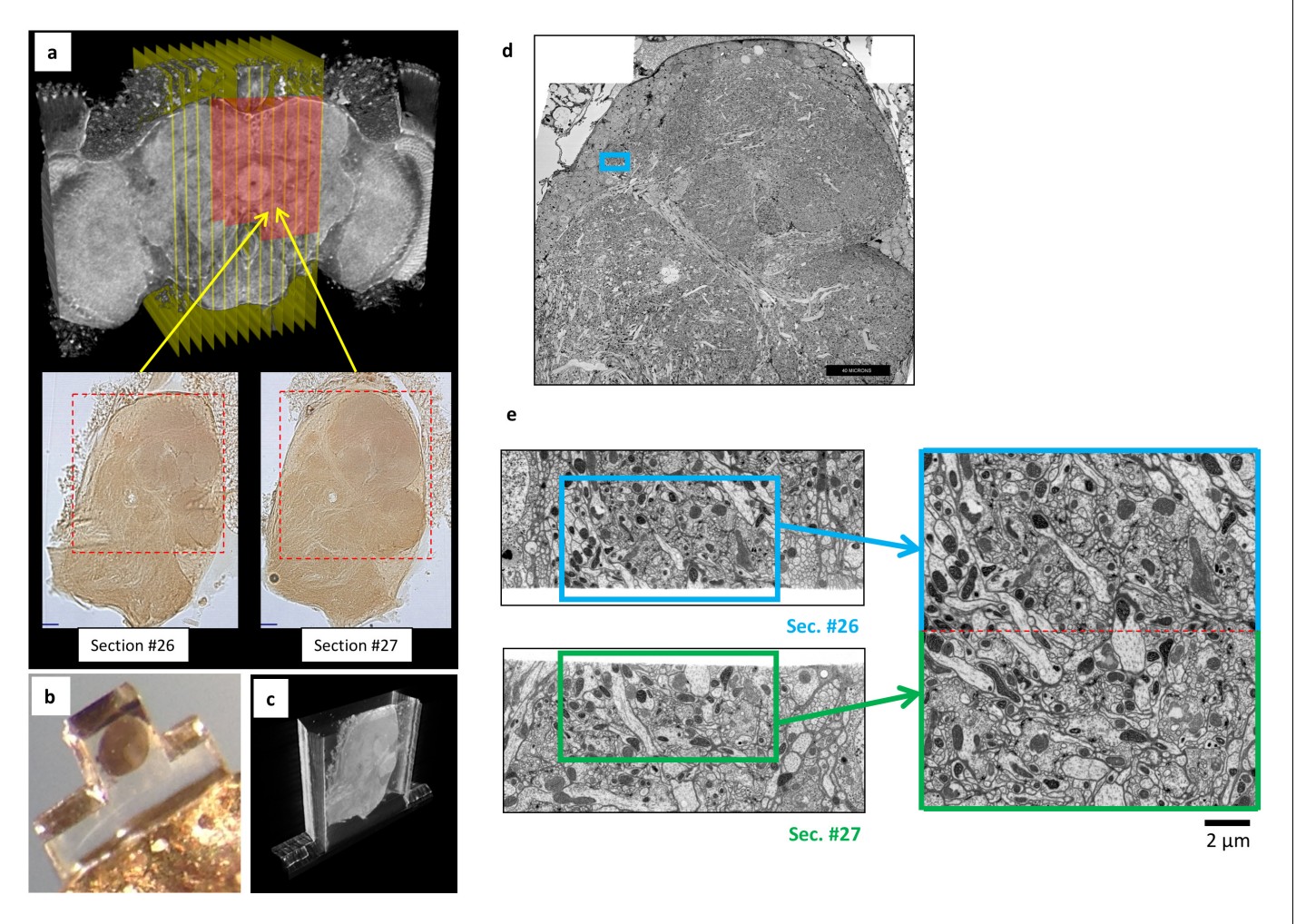

**Figure 5.** Overview of ultrathick partitioning and imaging results. (**a**) X-ray micro-CT of *Drosophila* brain cross-section shows central complex structures (the doughnut-shaped structure at center is the ellipsoid body). Yellow planes show locations of hot knife cuts at 20 µm intervals. Red highlighted area shows FIB-SEM imaged volume covering nine hot knife sections (#22-30 in our notation). Example light micrographs of Section #26 and #27 are shown (dashed box shows FIB-SEM imaged volume in each). (**b**) Each hot knife section is flat embedded against a PET laminate, individually mounted on a metal stud, and laser trimmed to dimensions suitable for efficient FIB-SEM imaging (*Hayworth et al., 2015*). (**c**) X-ray micro-CT of individually-mounted hot knife section showing laminar structure. All sections are micro-CT imaged as a quality control prior to FIB-SEM imaging. (**d**) Z-Reslice through FIB-SEM imaged volume of section #26. Blue box shows location of volume stitch test in protocerebral bridge region. Scale bar, 40 µm. (**e**) Result of volume stitch test in protocerebral bridge region. The FIB-SEM volumes of corresponding regions of adjacent hot knife sections #26 and #27 were computationally flattened and stitched to produce a single FIB-SEM volume suitable for tracing (*Hayworth et al., 2015*). Red dashed line shows stitch line. This stitched volume is available as *Video 8*. Scale bar, 2 µm.

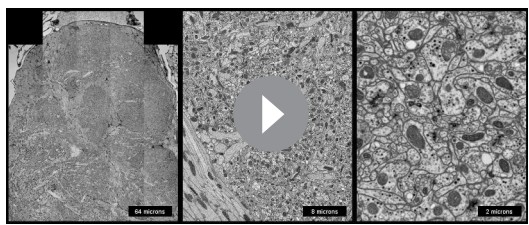

**Video 3.** Re-sliced views of a hot knife slab containing the *Drosophila* central complex at various zoom levels. The left panel shows the entire slab at 512 × 512 × 64 nm$^3$ voxel. The center panel shows a cropped region at the bottom of fan shape body (FB) with 64 × 64 × 64 nm$^3$ voxel. The right panel shows a cropped region in FB with 16 × 16 × 16 nm$^3$ voxel.

to keep up with the rate of data generation. The superior z resolution of the FIB-SEM data provides a more balanced pipeline for complete reconstruction of the fly brain with better-matched investment between acquisition and proofreading.

## FIB-SEM for high-resolution connectomics

This new capability for long-term operation opens up a whole new application space for high-resolution (~4 nm isotropic) 3D imaging. Exemplary data sets from *Drosophila* neuropil and *Chlamydomonas reinhardtii* illustrate the difference in resolution for high-volume vs. high-resolution acquisition (*Figure 6*). Considering the large processes found in mammalian neuropil, this may not be required for mammalian connectomics, but the additional resolution can be very useful to decipher the extremely fine processes in *Drosophila* neuropil (*Meinertzhagen, 2016*). The ability to explore synaptic motifs and other details of neuropil at high resolution can be very helpful in interpreting the larger volume but poorer resolution data sets required to generate a full connectome. The operating conditions needed, including lower current to reduce chromatic and spherical aberrations, and lower electron landing energy to reduce point spread function size along z-axis, require lower acquisition rates, implying smaller sampled volumes. The typical trade-off between resolution and volume for a given time constraint is illustrated by constant time contours in *Figure 1* (assuming our baseline *Drosophila* samples are used). The exact placement of these contours depends on specific features of the sample. For example, images can be acquired faster from mammalian neural tissue than from *Drosophila*, both because its stronger contrast and fewer small processes.

An 8 × 8 × 8 µm$^3$ high-resolution data set from the *Drosophila* central complex acquired over 10 days illustrates clearly delineated processes, and synapses with well-defined postsynaptic densities (see *Figure 7 and Video 4*). This high-quality data provides significantly more accurate estimates of synaptic connectivity than possible for lower-resolution stacks. Moreover, data of this quality will permit highly automated reconstruction, thus greatly reducing the time required for manual proofreading. In contrast, it would take only ~2 hr to acquire a comparable dataset using the high-throughput mode of acquisition, but it would require hundreds of man-hours of proofreading to correct the dataset because of the small neurites in *Drosophila*. High-resolution data also provides an accurate 'gold standard' for the higher throughput data, helping to interpret the larger dataset and perhaps also serving as a reference for machine learning. The actual resolution of this high-resolution mode can be objectively quantified by intracellular structures of known dimensions. For example, in *Figure 7b*, the resolved hollow core of the 25 nm outer-diameter/17 nm inner-diameter microtubule confirms a resolution of <3.5 nm (referenced to a 25–75% step edge rise resolution criteria, consistent with an alternative definition for resolution of (spatial period = 21 nm/2π).

## FIB-SEM for cell biology

Further improvement in resolution/volume for FIB-SEM should pay substantial dividends for cell biology, just as previous two-fold resolution improvements in fluorescence microscopy have enabled important scientific advances by rendering finer details. Here, we illustrate the potential of this approach for cellular neurobiology with data from the nucleus accumbens, a region of the mammalian forebrain involved in reward processing (see *Figure 8a* and *Video 5*). This dataset encompasses much of the soma of one neuron, along with the surrounding neuropil. The endoplasmic reticulum (ER) is well resolved and easily segmented, allowing its full 3D structure to be extracted; one is no longer relying on sampling from 2D EM sections to infer its 3D organization. For example, the frequency of ER-to-plasma membrane and ER-to-mitochondrion contacts can be quantified across a whole cell (*Wu et al., 2017*), providing new insight into contact-dependent processes such as lipid

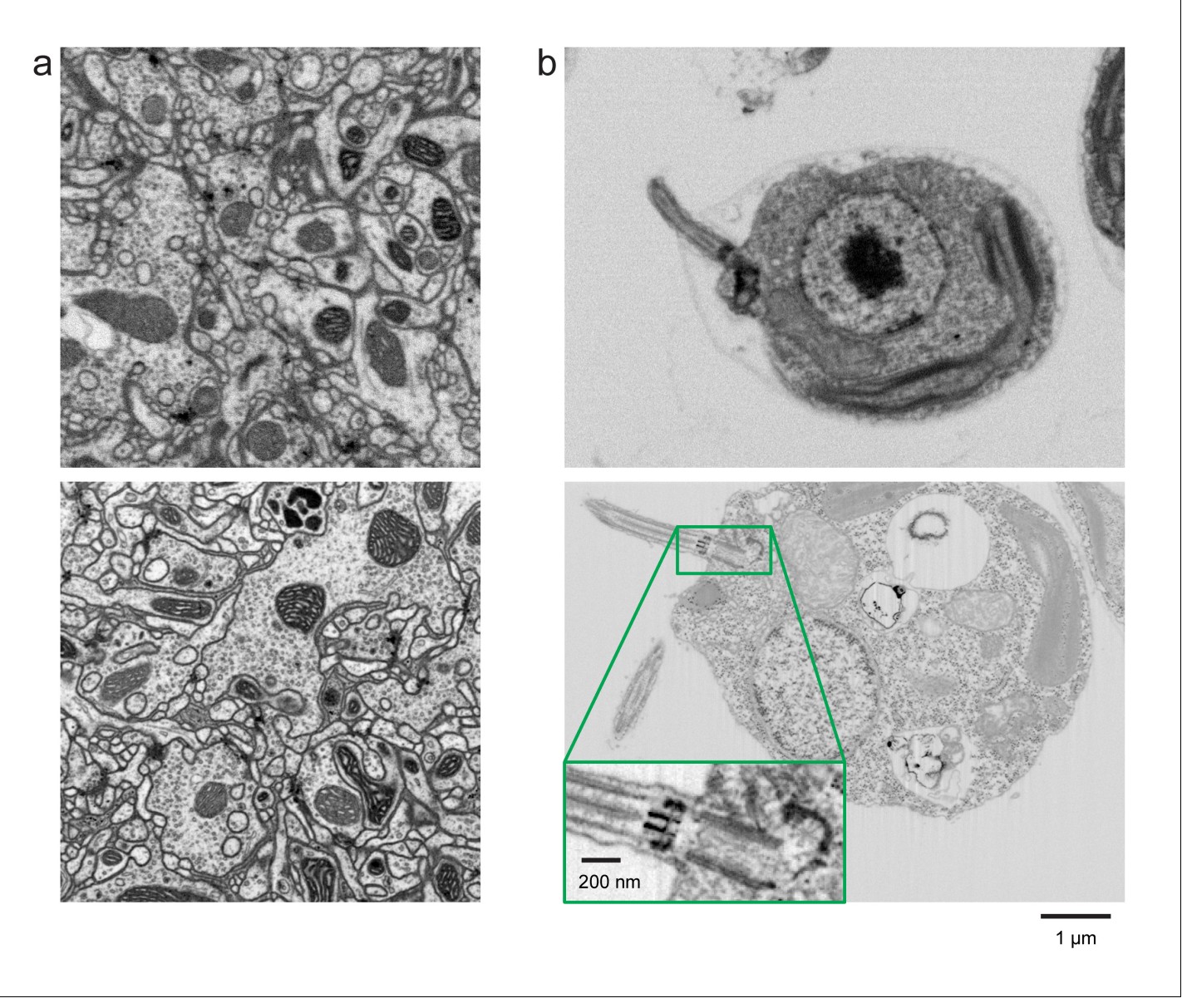

**Figure 6.** Improved FIB-SEM resolution reveals more detailed cellular structures in biological samples. Typical images of (**a**) *Drosophila* central complex and (**b**) *Chlamydomonas reinhardtii*, using standard $8 \times 8 \times 8$ nm$^3$ voxel imaging condition are shown in the top panels. The bottom panels show the corresponding high-resolution images at $4 \times 4 \times 4$ nm$^3$ voxel. Scale bar, 1 μm. Inset scale bar, 200 nm.

transfer. Dendrites with readily-segmentable organelles, and synapses with all vesicles countable are plentiful and could be mined for statistics, for example, quantitative comparisons among synapses from the same axon, or onto the same dendrite.

The ability to reorient the isotropic 3D data set provided by FIB-SEM permits high-resolution examination of arbitrary slices, thus offering new insights. For example, the nucleus accumbens volume reproduced here shows the edge of a soma, a partial nucleus, and the adjoining Golgi apparatus with a barely visible grey region associated (*Figure 8a*). By rotating that data block into a more informative plane (*Figure 8b*) one can see that this Golgi is corralled by a grey fibrous arc, a structure perhaps formed from rootletin (*Chen et al., 2015*), as suggested by the observed 120 nm periodicity. Remarkably, by following these fibers in 3D through multiple rendered image planes, we see that they connect and terminate onto the basal body of a cilium.

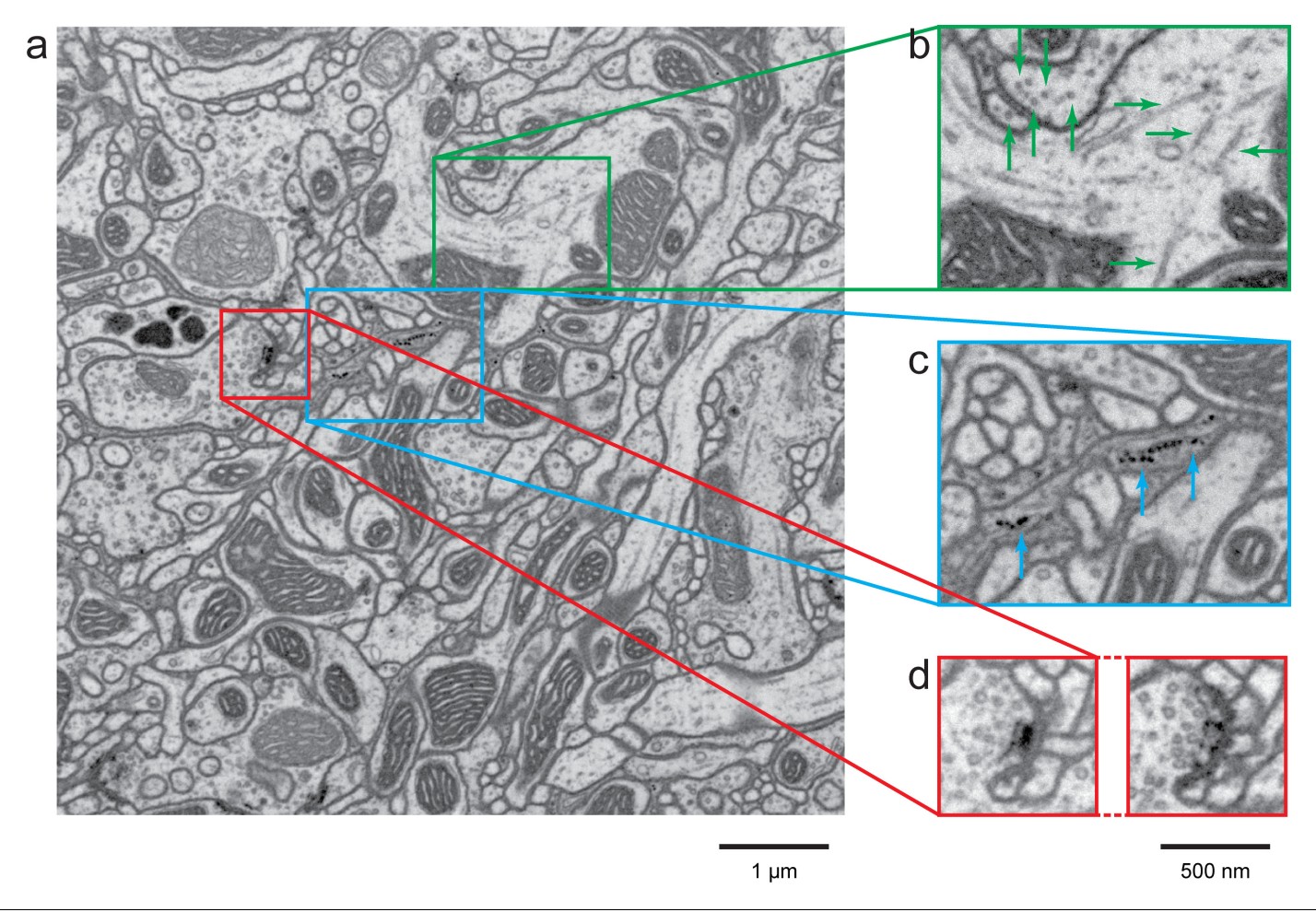

**Figure 7.** A high-resolution image (4 × 4 × 4 nm³) of a *Drosophila* protocerebral bridge (in the central complex) reveals fine details of various organelles. (a) an 8 × 8 μm² area overview; (b) end-on and side views of microtubule, indicated by green arrows; (c) polyribosomes attached to the endoplasmic reticulum, indicated by blue arrows; and (d) synaptic vesicles, presynaptic T-bar, and postsynaptic density, shown in two different z planes. *Video 4* shows the corresponding full z stack. Scale bar, 1 μm in (a) and 500 nm in (b)-(d).

In another demonstration of the potential of FIB-SEM for cell biology discovery, we imaged a group of the single-cell green alga *Chlamydomonas reinhardtii*. *C. reinhardtii* is widely used as model organism in the study of cilia/flagella structure and function, advancing our understanding of how defects in these organelles cause human diseases. It is also a broadly used model system for eukaryotic photosynthesis, chloroplast biogenesis, light perception, cell-cell recognition and cell cycle control (*Harris, 2001*). One cell cropped out of the data volume is shown in *Figure 6b*. Many details in the structure of the nucleus, mitochondria, ER, and Golgi are visible. Of particular interest is the large cup-shaped chloroplast and the associated light-sensing eyespot and the 'pyrenoid,' a Rubisco-rich structure involved in the first major step of carbon fixation. All these structures are clearly distinguishable; their overall organization and interplay, in various functional conditions and in light-sensitive mutants, would provide new light on mechanisms of photosynthesis.

*C. reinhardtii* flagella have also been extensively studied to understand cell motility. Zooming into the flagellar base (see inset *Figure 6b* and *Videos 6* and *7*), the nine-fold doublet microtubule structure becomes clearly visible, and details of the mature basal body pair and two probasal bodies are revealed. The latter form during basal body replication, at a very early stage of cell division in *C. reinhardtii*. After cytokinesis, the daughter cell will contain one mature basal body and one newly-formed one from which the flagella pair will grow (*Silflow and Lefebvre, 2001*; *Preble et al., 2000*).

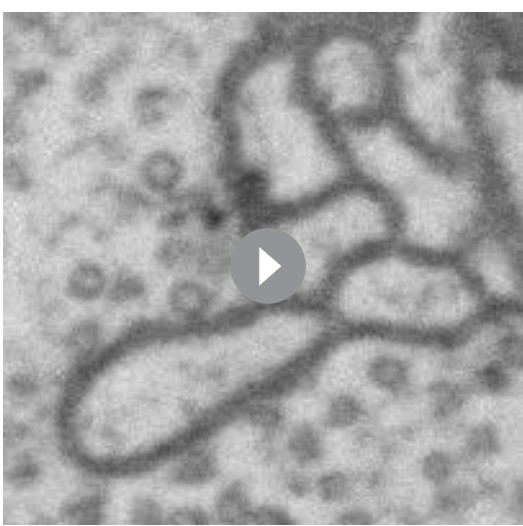

**Video 4.** Detail of synapse in *Drosophila* protocerebral bridge showing multiple post synaptic contacts.

Our imaging of this stage illustrates how recording a large data set that includes a population of cells allows the researcher to capture a variety of details through the cell cycle, as well as being more statistically meaningful than a single tomogram of a thin section of a single cell could ever be.

The wealth of structural data yielded by this approach merits the application of data mining tools. As an illustration of how further cellular details can be extracted from our FIB-SEM data set, we cropped out a thin spherical shell that starts at the nuclear envelope, extending 50 nm outward. By masking out backgrounds from inside the nucleus and those beyond 50 nm outside the nuclear envelope, we can restrict visualization to structures at or very close to the nuclear surface. This shell (rendered as a 3D section of a sphere in *Figure 8c*) reveals all the polyribosomes that decorate the exterior surface of the nucleus. Similar spirals of paired ribosome necklaces were seen previously in EM sections that happened to intersect the polyribosomes at just the right angle (*Christensen et al., 1987*). Here, by virtue of the isotropic character of the 3D data, one is no longer sampling a fortuitous section of the cell generated by the geometry of sectioning. Consequently, it is now possible to count all the polyribosomes on the whole nucleus. The insert of *Figure 8c* shows the nuclear pores, with their eightfold symmetry, in relation to these polyribosomes. Similar data can be extracted for ER-bound ribosomes throughout the cell.

## Quantification and optimization of FIB-SEM data acquisition

Most FIB-SEM images are assigned an arbitrary grey scale. This limitation deprives one of opportunities 1) to understand the true limits of SEM performance, 2) to optimize acquisition and identify inefficiencies, 3) to quantify absolute staining levels in samples, 4) to distinguish instrument vs sample factors, and 5) to cross-compare performance across different labs, samples, and FIB-SEM instruments. Initially, motivated by our desire to improve the speed of traditional SEM imaging, we needed to get a better understanding of the mechanisms by which backscattered electrons generate contrast, and to better define the ultimate limits of collection/detection efficiency. By comparing experimental results from specimens of known chemical compositions (gold, epoxy resin, and metal-organic compounds) with theoretical simulations using Monte Carlo methods for electron scattering (Figures 13, 15 and 17) and SIMION for electron optics (Figure 14), we characterized baselines under different sample biasing conditions (Figures 11 and 12). The general agreement between simulation and experimental results guided us to optimize SEM imaging with minimal artifacts (Figure 11). As a foundation for the experiments reported here, we also established two independent methods of quantifying the signal in terms of electrons detected. These explicit electron counts can be compared against models of electron scattering and also to reference standards, to establish best operating conditions (the results are detailed in the Technology and methods section).

## Imaging large volumes

A current milestone in connectomics is to image a 1 mm$^3$ volume. This is a daunting task for FIB-SEM, given its slower imaging speed compared to other competing methods. With the current throughput used for *Drosophila* brain, it would take approximately 100 years to acquire 1 mm$^3$ volume at $8 \times 8 \times 8$ nm$^3$ voxel resolution with a single system. However, the task might not be as hopeless as it seems. First, we have seen less demand for resolution in applications requiring large volumes. For example, mammalian brains have relatively larger processes and synapses compared to those of *Drosophila*. Mouse neuronal circuits should be traceable with minimal isotropic voxel of 16

**Figure 8.** Isotropic high-resolution data offer easy visualization of 3D structures through arbitrary slices. (**a**) Two orthoslices (x–y and x–z) from nucleus accumbens in a sample of adult mouse brain. (**b**) Specific slices of the same volume as in (**a**) provide easy viewing of Golgi in the context of other nearby organelles. Matching arrows in (**a**) and (**b**) indicate the same ROI's through different slice views, (**c**) Polyribosomes and nuclear pores (arrow) at the nuclear envelope of a *Chlamydomonas reinhardtii*. The 3D rendering was generated by thresholding a maximum intensity projection where brighter yellow (polyribosomes) indicates higher intensity of backscattered electrons due to stronger staining than darker yellow (nuclear pores). Scale bar 1 μm and 100 nm.

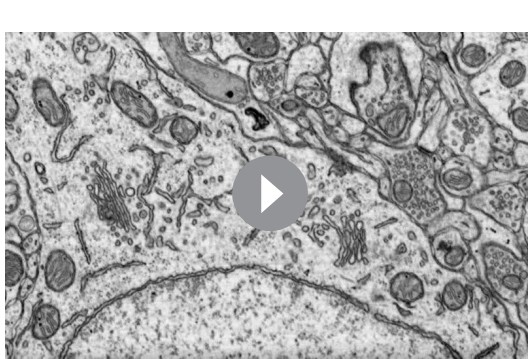

**Video 5.** Nucleus accumbens of a mouse brain.

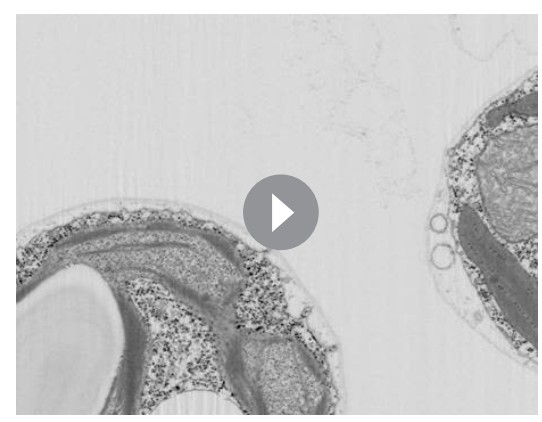

**Video 6.** Whole *Chlamydomonas reinhardtii*.

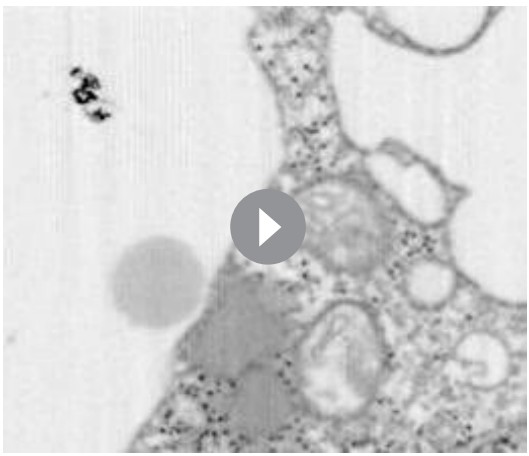

**Video 7.** Flagella structure of a *Chlamydomonas reinhardtii*.

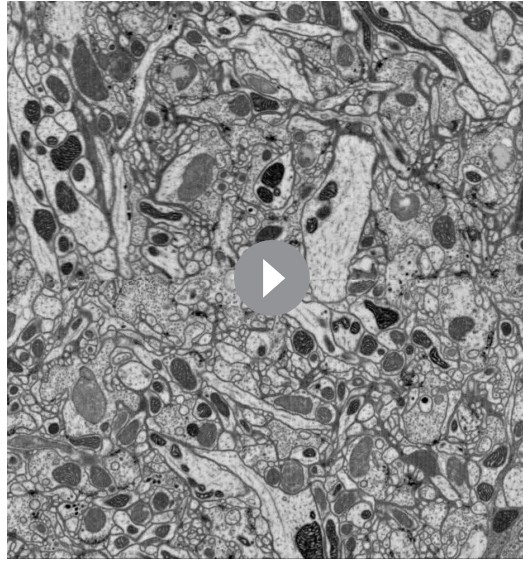

**Video 8.** Result of volume stitch test in *Drosophila* protocerebral bridge region between hot knife sections #26 and #27.

nm (***Mikula and Denk, 2015***). Secondly, protocols for mammalian sample preparation produce higher contrast than those for insects, which allows shorter imaging time. Being able to image with larger voxels benefits FIB-SEM volume throughput in two ways: fewer voxels (to the third power) and the ability to use larger imaging current, which allows shorter scanning dwell time to achieve the same shot noise, though the FIB milling overhead is increased. Based on limited data comparing *Drosophila* and mouse cortex samples acquired on the same FIB-SEM system, we estimate an 8x improvement on volume imaging speed with mammalian brain tissue. As illustrated in ***Figure 1***, one could reach 1 mm$^3$ with $16 \times 16 \times 16$ nm$^3$ voxels using a single FIB-SEM system in 8 years. Given that we can acquire samples from multiple systems running in parallel, one could expect a more feasible timeline: with our current capacity of four production systems, we estimate that a 1 mm$^3$ volume could be imaged in a total of 3 years, including hot-knife overhead and machine maintenance.

| Type | Interrupts | Frequency |
|---|---|---|
| Stability | System Drifting (nm to sub mm) | 1 day |
| Regular Maintenance | FIB Ga Source Reheat | 3-5 days |
| | FIB Ga Source Replacement | 3-4 months |
| | SEM Tip Replacement | 1 year |
| Facility | Room Temp Excursion | Few weeks |
| System Failure | System Component (e.g. Pump) Failure | Random |
| | Computer SW&HW Failure | 1 year |

| Challenges | Potential Impact to Image Data Set |
|---|---|
| Uneven FIB Milling | Irregular Z-step, Waves, Sample Damage |
| Loss of Static State | Out of Focus Images |

| Features | Benefits |
|---|---|
| Facility: Fault Tolerant Protection | Uninterrupted and Stable (e.g. Power, Temperature) |
| Auto Focus Auto Stig Auto Align | Maintain High Image Quality for Multiple Months |
| Prompt Pausing | Prevents Image Loss from Interrupts |
| Seamless Restart | Prevents Image Loss from Restart |

**Figure 9.** Major challenges to long-term reliability and stability of FIB-SEM systems. (**a**) System failure modes with different frequencies of occurrence and resume challenges. (**b**) Corresponding customized solutions.

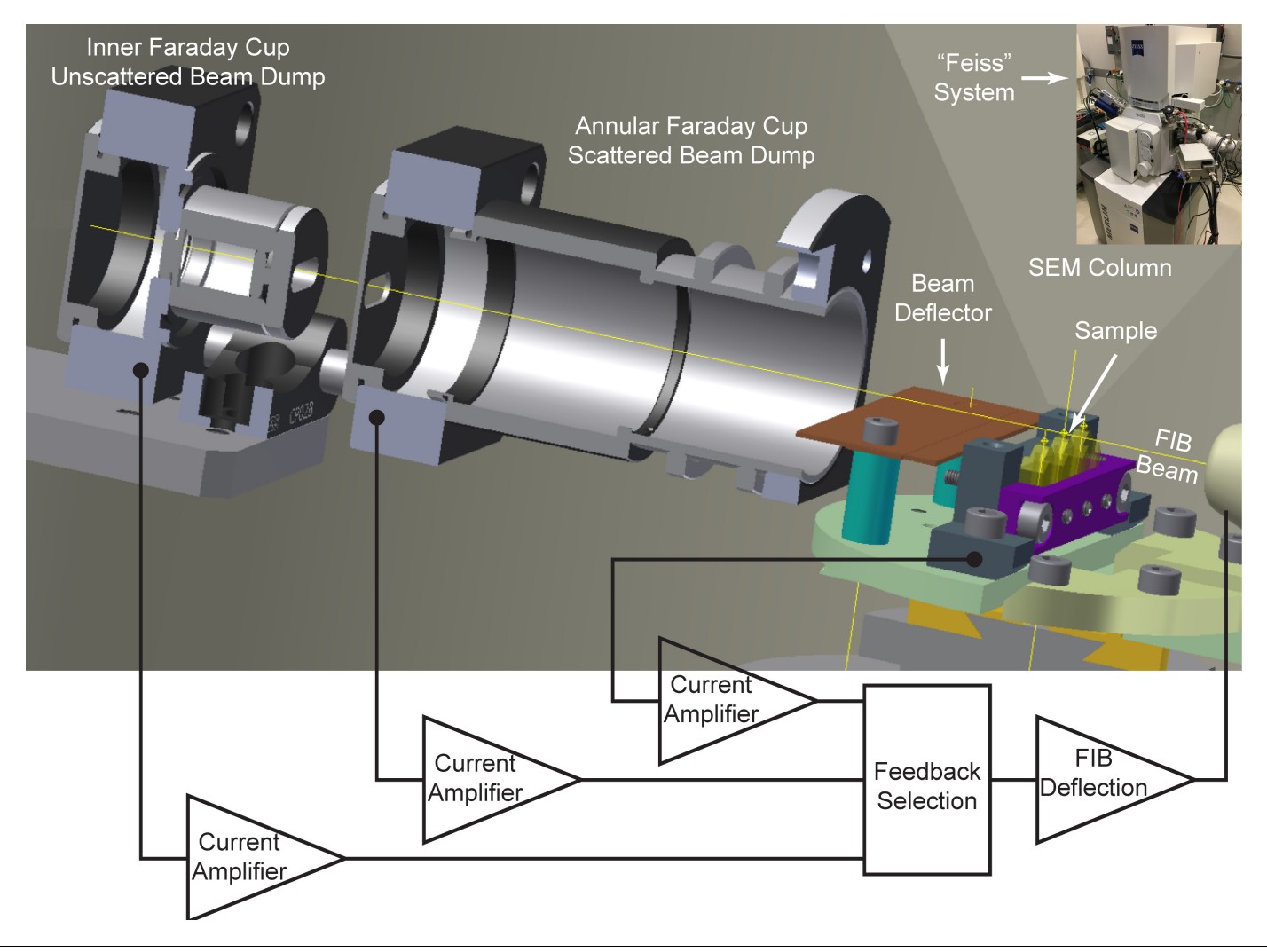

**Figure 10.** Cross-sectional diagram of closed-loop control set-up for FIB milling, showing the inner and annular Faraday cups. A beam deflector provides fine-tuning to steer the positively charged FIB beam into the inner Faraday cup. Feedback currents from specimen, annular, or inner Faraday cup can be used to control the FIB beam milling position for a targeted removal rate. Inset at upper right corner shows a picture of 'Feiss' system in which an FEI Magnum FIB column is mounted perpendicularly to the SEM column in a Zeiss Merlin SEM.

## Conclusion

The technical developments reported here now make it feasible to extend the intrinsic advantages of FIB-SEM, including excellent z resolution, isotropic voxels, and easy 3D data acquisition to larger volumes, by allowing long-term imaging for weeks, months, or even years. These enhancements can be adopted by other labs or on commercial systems to transform FIB-SEM into an effective tool for connectomics, which demands both high data quality and large data sets. The higher resolution mode of SEM imaging (~4 nm isotropic) can also be harnessed to study volumes of 5–50 μm linear dimension, providing a more detailed view of neuropil to guide the connectomics effort. For cell biology, the access to fine resolution and complete eukaryotic cell-sized volumes make this a practical alternative to the difficult and tedious stitched serial section tomographic TEM approaches now available. Thus, the technical advances reported here open new vistas for the study of biological structures.

# Technology and methods

## FIB-SEM limitations

SEM imaging is usually slower than that of transmission electron microscopy (TEM) for several reasons. First, SEM acquires images pixel-by-pixel in series, whereas TEM acquires all pixels of the image in parallel, with orders of magnitude larger imaging current. Second, SEM detects backscattered or secondary electrons, emitted at a much smaller flux than the transmitted electrons measured by TEM. Third, the low SEM landing energy of <2 keV typically used to reduce the electron penetration depth into the block-face (thus increasing z-axis resolution) can reduce contrast, especially below 800 eV (Figure 15). Fourth, the signal-to-noise ratio (SNR) of a pixel depends on the number of primary electrons devoted to it, which in turn is determined by the beam current. For resolution of <10 nm at low landing energy, the incoming electron beam current must be limited to <10 nA, since higher current beams require larger apertures, which are subject to greater spherical and chromatic aberrations as well as Coulomb repulsion, creating unacceptable blur in the beam spot. Finally, post-staining is commonly used in TEM and ATUM-SEM (*Hayworth et al., 2006*) sections to enhance contrast, whereas FIB-SEM must image the block surface without benefit of the extra contrast from post-staining. As a result of all these factors, FIB-SEM and other block-face scanning methods (e.g. serial block-face scanning electron microscopy) require a lower image acquisition rate than TEM to achieve the same SNR.

Along with slow throughput, the limited duration of continuous FIB-SEM data acquisition constrains the useable volume. FIB-SEM is destructive and does not allow re-imaging, imposing formidable requirements on system reliability. Interrupts have a direct impact on the total contiguous imaging volume. System drift, routine maintenance, facilities interrupts, or system failures can all terminate a 3D FIB-SEM operation. Focus or beam stigmation of SEM can drift from its optimal settings within 1–2 days, due to environmental or sample stage instability. A pause in the milling/imaging operation is normally needed to correct these drifts and restore image quality. Moreover, a FIB gallium source has a lifetime limited to 3–4 months of continuous operation, and requires reheat or 'flashing' every 3 to 5 days. Even without facility or system failures, these regular maintenance events impose hard limits on continuous data acquisition, and thereby the size of contiguous high-quality data sets that can be collected with standard FIB-SEM systems.

## FIB-SEM system customization for continuous long-term acquisition

The ability to operate the system for long periods is crucial for imaging large volumes. Unfortunately, there are many potential interruptions to a FIB-SEM system, with intervals ranging from a few hours to a year (*Figure 9*). Some of them relate to system reliability, which is challenging to improve. Others are regular maintenance items that are impossible to eliminate, such as FIB source reheat (3 days) and replacement (3 months). Since FIB-SEM image acquisition is destructive, any interrupt could be detrimental. For example, a spike in room temperature may cause the SEM focus to drift, and the FIB beam-pointing position relative to the specimen to change, potentially damaging the sample and sabotaging the continuity required for neural tracing of fine processes across the brain. To address these frequent interrupts, we have developed a system that immediately pauses to prevent damage, and resumes seamlessly after restoration of normal operation. By providing high virtual reliability, this system greatly expands the total imaging volume possible.

### Facility

Fault-tolerant protected facilities with layers of backups were implemented on all utilities to provide uninterrupted and stable power, cooling water, compressed air, room temperature, air handling, etc. These back-up utilities were further tuned so that any transition would not produce a damaging fluctuation. For example, switching from a failed main air handler to an auxiliary unit will produce a transient in room temperature of <0.5°C. Although costly, these facility upgrades are the foundation of stable long-term operation. A second layer of protection is achieved by monitoring environmental parameters like room temperature, so that the system can be paused quickly in the event of an anomalous excursion.

## In-line image auto-optimization

During extended image acquisition, SEM focus, stigmation, and aperture alignment must be optimized periodically to correct drift of various components. Traditionally, these adjustments are performed manually off-line, causing interrupts to the continuous milling and imaging process. Besides increasing overhead, this approach introduces further errors by disturbing the steady state of milling and imaging cycles. The in-line optimization procedure described below effectively overcomes the deficiencies in conventional methods. Recognizing these problems, a functionally similar approach was previously described (*Binding et al., 2013*).

In our approach, a 'focus index' (*FI*) based on the strength of the highest spatial frequency components is calculated to assess the focus quality for each acquired image. $FI = [\sum_{i=0}^{n-1} \sqrt{(I * S_1 - I * S_2)^2}]/n$, where the original 2D image *I* is smoothed by 2D Gaussian functions of a shorter ($S_1$) and a longer ($S_2$) length scale that straddles the resolution limit. Typical standard deviation values for Gaussian distributions $S_1$ and $S_2$ are 1 and 4 pixels, respectively. The total root mean square pixel-to-pixel difference between those two smoothed images averaged over the total number of pixels (*n*) is assigned to the original image as a focus index, where *i* is the index of each pixel. A higher focus index indicates more high spatial frequency content, and therefore a better-focused image. The focus index value depends upon the highest spatial frequency features of the sample, the actual focus of the electron beam spot, and the signal intensity. When comparing neighboring frames in a 3D FIB-SEM image stack, there proved to be only minimal variations in either sample features or signal intensity, so focus index is conveniently sensitive to the beam spot size. Alternatively, the *FI* can be obtained by other methods (*Binding et al., 2013*) that quantify the 'sharpness' of an image.

To initiate an in-line auto focus procedure, a series of SEM images are taken (as part of the ongoing FIB-SEM acquisition) 1–2 μm over- and under-focus from the current value. A parabolic curve fit to the focus index vs. defocus is used to extract the optimal focus setting, maximizing the focus index. Upper and lower bounds of the optimal setting are specified to prevent outliers. This derived optimal setting is then applied to subsequent images (correcting for the anticipated z removal from milling that is incorporated into the target focus setting). Using the same basic approach, stigmation and aperture alignment settings for x and y axes are also optimized. After the system reaches steady state for milling and imaging (typically within 1–2 hr), a few iterations of the auto-focus, stigmation, and alignment routine are used to optimize SEM imaging condition. To continually correct subsequent slow drifts of the system, the routine is automatically triggered by software every 200 frames or every few hours. This infrequent sampling and almost imperceptible defocus minimizes any possible compromise in the data quality. Importantly, it does not introduce any throughput overhead or additional radiation damage.

## Closed-loop control for FIB milling

The precision sectioning process is the most unforgiving component of FIB-SEM. Loss of control of the focused ion beam can destructively ablate as much as a full micrometer of material without the associated imaging, thereby destroying the continuity of large data sets with potential loss of months of invested imaging effort. Even small instabilities can nucleate waves and curtains of non-uniform milling. Beyond ensuring stable ion beam parameters such as current, stigmation, and focus as intrinsic aspects of the ion column, to ensure reliability, we found it necessary to provide a feedback mechanism to regulate the ion beam milling height.

A previously described feedback scheme (*Boergens and Denk, 2013*) captures the part of the 30 keV focused ion beam that does not hit the sample in a Faraday cup. This non-occluded beam current is measured and subtracted from the total beam current as measured in the FIB column Faraday cup, providing an estimate of the total beam current impinging on the sample. An increase in this impinging current raises the beam (to lessen the occlusion), whereas a decrease in beam current lowers the beam into the sample, thereby increasing milling while reducing the Faraday cup current. We use a modified version of this concept to give two further monitored values and feedback options, as diagrammed in *Figure 10*. An inner Faraday cup captures the non-scattered ion beam, but with a smaller acceptance slot consistent with the beam spread and horizontal scanning. A second annular Faraday cup captures the more widely scattered ions and milled sample material, providing a more direct measure of milling rate. Finally, a current is generated on the sample by the milling beam,

which should be proportional to the milling rate. A current preamplifier that could be biased to ±800 volts enabled measurements of these currents under a wide range of sample bias voltages.

All three modes were tested for feedback control of milling; each exhibited strengths and weaknesses. If feedback is based on the inner Faraday cup, operation is simple, but variations in the ion source emission properties such as beam shape are not compensated and can lead to a non-uniform milling rate. To some extent, this can be corrected with post-processing software to normalize an estimated milling (z) increment (*Hanslovsky et al., 2014*). The sample current is a composite of multiple components, including not only the impinging positive milling ions, but also secondary electrons, as well as any charge that the milled sample atoms remove. As a result, the value and sign of the feedback is poorly determined, and under certain beam shape condition, it can be even close to zero, potentially leading to milling instabilities. A third option–feeding back on a larger annular Faraday cup that captures scattered $Ga^+$ ions and any charged ions milled from the sample–yields the most uniform milling rate, displaying only minimal sensitivity to changes in the gallium source emission profile and ion currents. We find it the preferred mode of operation, but also use bounded checks on the other currents to allow continuation of milling.

## Prompt pausing and seamless restart

To foresee upcoming interrupts and react proactively, the health of the FIB-SEM system is monitored in real-time, recording machine and environment parameters, to generate trend charts. Standard statistical process control methods are used to construct control bands of these parameters (e.g. 3x the 25–75 percentile range for the trailing 100 data points) in real-time. In the event of any excursions beyond control band limits, control software alerts the operator through email or (depending on its magnitude) automatically pauses the operation. Parameters monitored include room temperature, specimen current during FIB milling and SEM imaging, FIB beam position, image X-Y shift, image focus index, and feature change between adjacent frames, etc.

A prompt pause of the system prevents damage to the specimen, but it proved challenging to resume the operation seamlessly after system restoration. The most difficult problem was how to re-aim the FIB beam back to the position just before the pause with nanometer precision, to avoid over or under-milling. In addition, during repeat cycles of milling and imaging, steady states of charge and temperature had been established on the block-face that required specific focus and stigmation settings for the SEM. These imaging parameters were optimized for the steady state, and thus would not be suitable for a cold start. Prior to our implementation of closed-loop FIB beam positioning, these two effects led to a large number of suboptimal images. It was not uncommon to perform tens or even hundreds of milling and imaging cycles before restoration of the steady state. This could lead to a gap as wide as 100 nm in the image stack, which would be devastating for tracing neural circuits. Besides instability during re-engaging from a cold start, material removal in each milling cycle often fluctuated due to various instabilities of the system.

To ensure correct milling rate in real time, we utilized the closed-loop control of FIB described above. The ability to detect and control the positioning of the FIB beam relative to the sample block-face with no delay and overhead effectively eliminated the uncertainty for FIB re-engagement, ensuring consistent milling at all times. Furthermore, we found that when the system deviated from steady state, the SEM focus and stigmation changes correlated nicely with the lateral image shifts, which were easily obtained by registering adjacent images. Standard image registration methods such as cross correlation, Scale Invariant Feature Transformation (SIFT), or Super Robust Feature Transformation (SRFT) could be used to calculate image shifts between adjacent frames. An empirical model was developed to apply offsets derived from image shifts to the optimal imaging parameters until the steady state was restored. The time to reach steady state was significantly shortened with the aid of a shutter underneath the SEM column (see Section "FIB-SEM column configuration"). We found that in most cases, no offset was needed after the SEM shutter was implemented. The FIB focus was also monitored before and after gallium source reheat, and controlled for smooth continuation. By implementing these measures, image focus could be restored immediately or within one frame after a restart, eliminating seams in the final image stack.

## Multiple magnifications at user-selected intervals

Multiple resolutions are often needed to resolve different features in an image stack, and to provide context. One option is to image the entire volume using the highest resolution needed. However, it proved far more efficient to change imaging resolution at different locations. For example, in a sample of fly brain, small and dense neuropil structures are surrounded by large cell bodies. Instead of imaging everything at high resolution, only the smaller regions of interest containing the neuronal connections are imaged at high resolution, while the larger area image(s) could be acquired more quickly at lower x, y, and z resolutions. These ROI's of different sizes and locations are first defined in the control software by boxes overlaid on a low-magnification image. The acquisition intervals of each ROI are also specified. Pre-programed gradual lateral shifts of these ROI's are often needed to keep the features of interest within the imaging boundaries.

## FIB-SEM column configuration

In most commercially available FIB-SEM systems, the SEM and FIB columns are mounted at an angle of 52–55 degrees. This versatile configuration allows a wide range of applications but is suboptimal for 3D volume imaging. Because the milled block-face is not perpendicular to the SEM column, the focus (also known as working distance) of the scanning electron beam needs to be dynamically adjusted as it scans across the tilted surface; this not only affects the SEM image quality near the top and the bottom edges but also constrains the flexibility of rotating a typical rectangular scanning area to accommodate sample shapes. Moreover, the coincident point between the SEM and FIB beam is limited to ~5 mm down from the bottom of SEM column, due to space constraints of the two columns. The resulting long working distance reduces the signal, due to the smaller solid of collection (especially using the InLens detection scheme [*Weimer and Drexel, 2002*]), and degrades the resolution, due to the poorer focus at longer working distance. To overcome these difficulties, we mounted an FEI Magnum FIB column perpendicular to the SEM column, onto a Zeiss Sigma or Merlin SEM system. This combined system (which we term 'Feiss') permitted a working distance of <3 mm, producing superior SEM images with greater flexibility. We found that the column mounting orientation affects the Magnum FIB emission characteristics, and closed-loop control is required to ensure milling stability. The bottom of the SEM column was prone to contamination by FIB-sputtered material over time due to its close proximity, reducing image quality and affecting system stability especially after a system pause. Contamination of the SEM column from FIB-sputtered material was further accelerated with shorter working distance. Accordingly, a mechanical shutter was installed to shield the SEM column during FIB milling. The shutter kept the bottom of SEM column debris-free even after hundreds of thousands of milling cycles, and also greatly reduced focus and stigmation drift when the FIB re-engaged. This shutter arrangement would be even more valuable if a beneath-the-lens electron detector were used.

## Signal quantification

A quantified SEM image signal whose grey-scale values correspond to a known number of detected electrons is an important aid to understand the properties of the SEM detector and the staining level of a sample. To achieve this quantification for any specific brightness and contrast setting of the SEM, a background image is first collected by blanking the electron beam, defining the grey value corresponding to zero electrons. The response of the SEM detector can then be calibrated using two independent methods. The first method relies on a 'total reflection mode,' in which a spherical specimen (a 3-mm gold-coated ball) is biased to a negative voltage equal to or higher than the energy of the incoming electron beam, sending the full beam into the detector. This primary beam current is independently calibrated with a Faraday cup. The second method uses a shot noise estimate, based on the Poisson distribution of electron counts around a mean. However, the shot noise measurement contains contributions from other noise sources, including the photomultiplier tube (1 + 1/gain) and the amplifier (1/f noise). The bandwidth limitation of the amplifier may also skew the outcome at fast scanning speeds, and the entire measurement depends on the sample. Therefore, we prefer to use the total reflection mode approach. With detector signal intensities calibrated to the true number of detected electrons, specimen qualities such as the overall staining and contrast can be quantitatively monitored and compared.

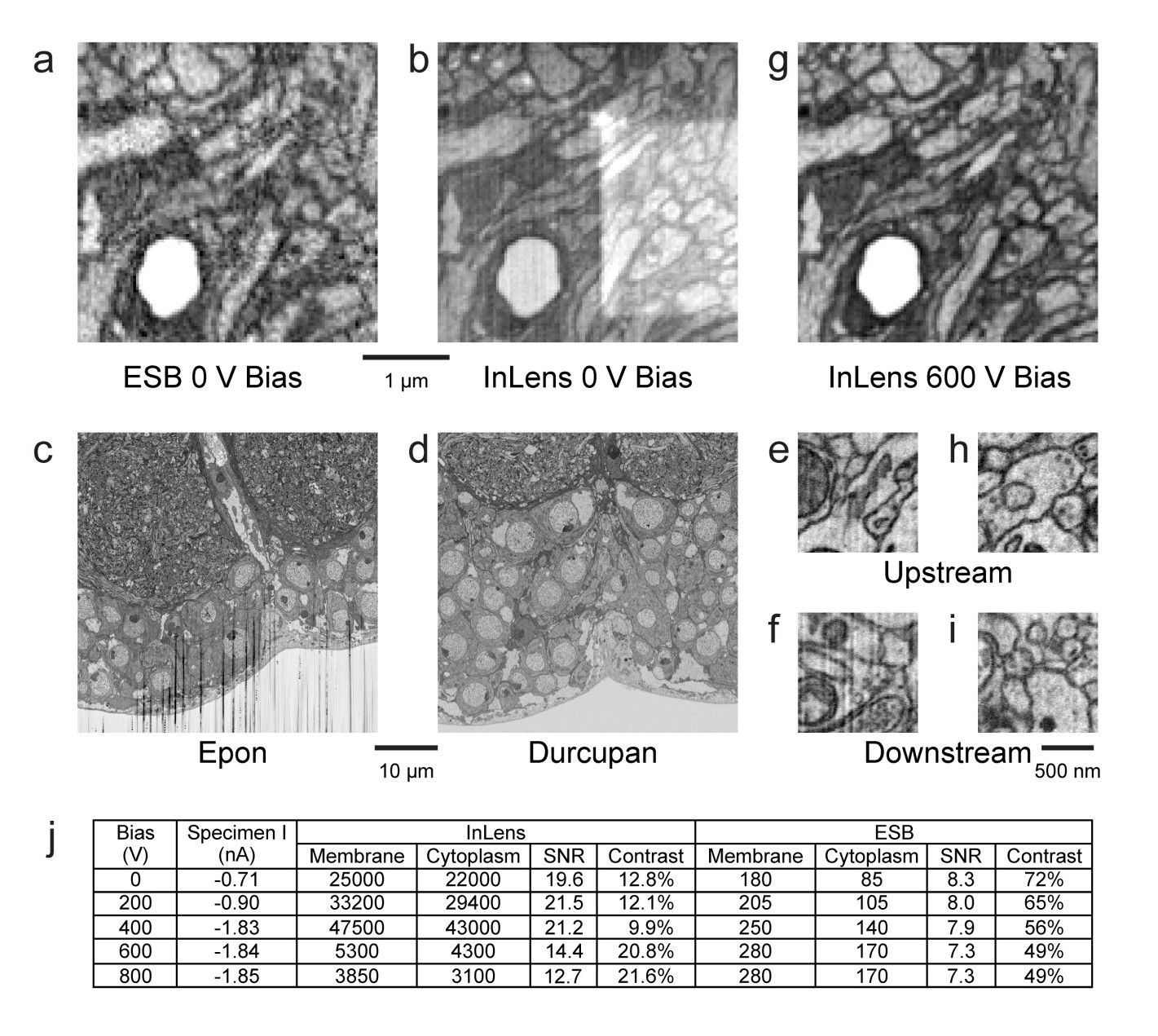

| Bias (V) | Specimen I (nA) | InLens | | | | ESB | | | |
|---|---|---|---|---|---|---|---|---|---|
| | | Membrane | Cytoplasm | SNR | Contrast | Membrane | Cytoplasm | SNR | Contrast |
| 0 | -0.71 | 25000 | 22000 | 19.6 | 12.8% | 180 | 85 | 8.3 | 72% |
| 200 | -0.90 | 33200 | 29400 | 21.5 | 12.1% | 205 | 105 | 8.0 | 65% |
| 400 | -1.83 | 47500 | 43000 | 21.2 | 9.9% | 250 | 140 | 7.9 | 56% |
| 600 | -1.84 | 5300 | 4300 | 14.4 | 20.8% | 280 | 170 | 7.3 | 49% |
| 800 | -1.85 | 3850 | 3100 | 12.7 | 21.6% | 280 | 170 | 7.3 | 49% |

**Figure 11.** Images of ultra-thin section of *Drosophila* brain on silicon substrate highlight the advantages of our specimen bias scheme.  (a) EsB signals had higher contrast but lower signal-to-noise ratio (SNR) compared to InLens (b). With 600 V bias (g), InLens contrast substantially increased, with only a small drop in SNR. In addition, artifacts such as electron burn marks were eliminated by the positive bias. Streak artifacts depend on embedding resin; Epon (c) is far less satisfactory than Durcupan (d). Streak artifacts are less prominent upstream (e,h) and more prominent downstream of the ion milling (f). A 600 V bias can eliminate the surface topography contrast that makes these streaks visible (i). (j) Specimen bias effects are quantified through SNR and contrast. SNR was calculated as $(N_m - N_c)/sqrt((N_m + N_c)/2)$, where $N_m$ and $N_c$ are electron counts of membrane and cytoplasm respectively. Contrast was calculated as $(N_m - N_c)/((N_m + N_c)/2)$. Scale bar, 1 µm in (a), (b) and (g), 10 µm in (c) and (d), 500 nm in (e), (f) (h), and (i).

## SEM artifact reduction

The detected electrons include both energetic backscattered electrons and lower energy but more numerous secondary electrons. Because the backscattered detector (*Figure 11a*) provides clean and excellent material contrast, it is the de facto choice for FIB-SEM applications. In the Zeiss Gemini SEM column (*Weimer and Drexel, 2002*), each signal can be detected separately with an EsB and

an InLens detector. A larger solid angle of the secondary electrons can be collected by the InLens detector, allowing faster imaging. Unfortunately, the secondary electrons contain new noise terms and artifacts arising from charging, topography, burn marks, and non-uniform work function (*Figure 11b*).

Some of these artifacts are sensitive to the material used. For example Epon, a common epoxy embedding compound, develops streaks and uneven milling especially downstream (in the milling direction) on the milled surface (*Figure 11c*). Alternate epoxy embedding resins such as Durcupan are much less prone to such streaks (*Figure 11d*). With the latter resin, streaks are absent at the top of the image (*Figure 11e*) but can develop 20 μm (*Figure 11f*) downstream in the milling direction when imaging with a dose >50 electrons per nm$^3$. If the streaks are sufficiently mild and occupy only a small part of Fourier space, they can be removed by applying a masked Fourier filter that removes the spatial frequencies of the streaks.

We found that a positive bias of the specimen above 500 eV can more directly and effectively filter out low-energy secondary electrons that are detected by the InLens detector, while maintaining a larger collection angle for the backscattered electrons (*Figure 11g*). The positively biased InLens image eliminates the rectangular electron burn spot and appears very similar to that of EsB image (*Figure 11a*), but with noticeably improved SNR. The bias enables the InLens detector to offer much improved material contrast, with a ~5–10x gain on electron counts compared to EsB alone, and also removes the mild streak artifacts otherwise visible ~20 μm downstream on the milled surface (*Figure 11i*). Moreover, the signal from both detectors can be combined through a simple weighted average to further lower shot noise without degrading image contrast (*Unser and Eden, 1990*).

## SEM signal-to-noise ratio

SEM imaging is usually the rate-determining step in a FIB-SEM procedure, since the milling procedure is faster (typically up to a voxel size of 24 nm or greater). To understand the limits of fast imaging, it is important to consider the SNR of the image. The signal is simply the number of electrons that backscatter into a detector from an osmium or other heavy-metal-rich stained membrane, $N_m$, minus those that scattered by the unstained cytosolic region, $N_c$. (both $N_m$ and $N_c$ are signals in a pixel of an electron micrograph). These numbers scale with the number of primary beam electrons, $N_p$ impinging on the sample. In our case, since the signal of the stained membrane over unstained cytoplasm $(N_m - N_c)$ is much less than $(N_m + N_c)$, the average number of detected electrons $N_{e^-}$ is approximately $(N_m + N_c)/2$. The noise as determined by statistics of the limited number of electrons collected with one pixel is given by sqrt($N_{e^-}$). The ratio of signal (of membrane over cytoplasm) to average image noise, SNR, can be approximated as: $SNR=(N_m - N_c)/sqrt(N_{e^-})=(N_m - N_c)/sqrt((N_m + N_c)/2)$, which is proportional to the square root of the number of electrons associated with the pixel. This in turn scales with the primary beam current, $I_p$, times the dwell time t of a pixel, $N_p = I_p *t$.

## Sample bias

Using an ultra-thin section of *Drosophila* brain on a silicon substrate (*Figure 11a*), we quantified the image quality for different detectors and different specimen bias voltages. We tested bias voltages ranging from 0 to 800 V, while adjusting electron beam energy accordingly to maintain a fixed landing energy of 1.2 keV. The distance between specimen and electron column was fixed at 3 mm. Electron counts from cell membrane and cytoplasm regions over a 2 μs sampling period for each pixel were obtained based on the detector calibration. SNR and contrast for each condition was calculated. The table of *Figure 11j* summarizes the typical low electron count but excellent contrast for EsB detection and the lower contrast but larger signal for wider angle InLens detection at various bias voltages.

The total electron counts with the InLens detector increased as a function of bias voltage, reaching a broad maximum between 0 and +400 V before a sharp drop-off at 600 V and beyond. This initial increase of collected electrons was likely due to a lensing effect near the column entrance that allows more electrons to reach the ring-shaped InLens detector. The dramatic decrease of electron signal with bias voltage above +600 V indicated the threshold of secondary electron removal. This threshold bias is consistent with the formation of a ~ 50 eV axial potential barrier that is formed by proximity to an +8 kV electrode together with a grounded end cap and thus capable of blocking secondary electrons to the InLens detector (Figure 14). As expected, the threshold is a function of

the distance between specimen and electron column. A shorter working distance requires higher bias voltage to effectively filter out secondary electrons. The main advantage of the EsB detector is the high contrast of ~40% to 50%, compared to ~10% provided by the InLens detector. However, the total number of electrons detected by EsB is only a small fraction ($\leq$1%) of those recorded by the InLens detector. Adding a bias voltage of +600 V nearly doubled the contrast of the InLens signal up to around 20%, while the SNR stayed above 10, about twice of that provided by EsB.

Next, we sought to quantify mammalian neuropil that was undergoing the cyclical FIB-SEM imaging and milling. The artifacts are minimized under stable FIB-SEM milling conditions, allowing other bias options for higher throughput imaging (*Figure 12*). *Figure 12a* show a gradual increase of electron counts with bias voltage stepping up from −600 to 400 V. In all cases, the landing energy is 1.2 keV. The sudden drop in electron counts at 500 V bias is consistent with the thin section results mentioned above, indicating the threshold of secondary electron removal. The values for the membrane and cytosolic signal $N_m$ and $N_c$ are empirically determined from the grey scale images (*Figure 12c* and *Figure 12—figure supplement 1*) to represent an average membrane grey level and an average cytosolic grey level. This is a somewhat subjective manual operation: A threshold red level is adjusted by eye until the membranes are about 50% covered by red, to get the representative median membrane grey value. Likewise, a green threshold is adjusted to get the median grey value of the empty cytosol (*Figure 12d* and *Figure 12—figure supplement 1*).

The grey scale calibration allows one to translate the red and green grey scale thresholds to electron signals $N_m$ and $N_c$, respectively. These values are also indicated as short vertical lines on the signal histograms in *Figure 12e* and *Figure 12—figure supplement 1*. Several quantities are tabulated for these images generated with a primary beam of 12,400 electrons per pixel, *Figure 12f*. The average number of detected electrons $N_{e^-}$ is approximately $(N_m + N_c)/2$, and the signal strength $N_m$ - $N_c$, defines the measured contrast $(N_m - N_c)/N_{e^-}$. The noise is approximately given by the shot noise, proportional to the sqrt($N_{e^-}$). An independent noise estimate can be derived from the pixel fluctuations (data – nearest neighbor smoothed data) This assumes that high spatial frequency components of the sample are not dominant in the nearest neighbor pixel-to-pixel variation, which is instead dominated by sampling fluctuations. This signal to noise ratio, SNR, is also tabulated. The comparison, although not rigorous, demonstrates the changes in SNR as a function of sample bias voltage. In addition, it serves as an experimental reference for Monte Carlo simulations in the following section.

We conclude that positive specimen bias provides a simple and effective alternative to generate strong EsB-like material contrast images using the built-in InLens detector in the Zeiss Gemini platform. The resulting 'artifact-free' images have sufficient contrast and SNR for high-quality automatic segmentation of neural tissue. With this approach, we were able to boost the throughput by a factor of 10 or more over EsB detection alone. However, when the steady state FIB-SEM imaging generated only limited and tolerable artifacts (e.g. streak artifacts, which could be removed by a simple mask on the Fourier transform), the 0 to +400 V sample bias and the larger aperture of the InLens detector gave the best signal-to-noise and throughput performance.

## Signal and resolution: comparison to ion optic and Monte Carlo simulations

To gain more insight into mechanisms underlying the detection efficiency, its limits and imaging resolution, we modeled the physics of the electron/sample interaction and the scattered electron detection efficiency, validating the models by comparison with known 'calibration' samples. Only a small fraction of the electrons from the incoming primary electron beam will be scattered back and collected by the detector, and an even smaller variation of that signal corresponds to the contrast generated by the stained and unstained parts of the sample. Monte Carlo simulations can provide a useful perspective. The approach described by David Joy (*Joy, 1991*) was adopted, using the Rutherford energy loss formula between scattering events, modified to use the angle- and energy-dependent elastic scattering cross-sections for the key elements H, C, N, O, P and Os. All these scattering cross-sections for all energies were obtained from a NIST database (*Jablonski A et al., 2016*).

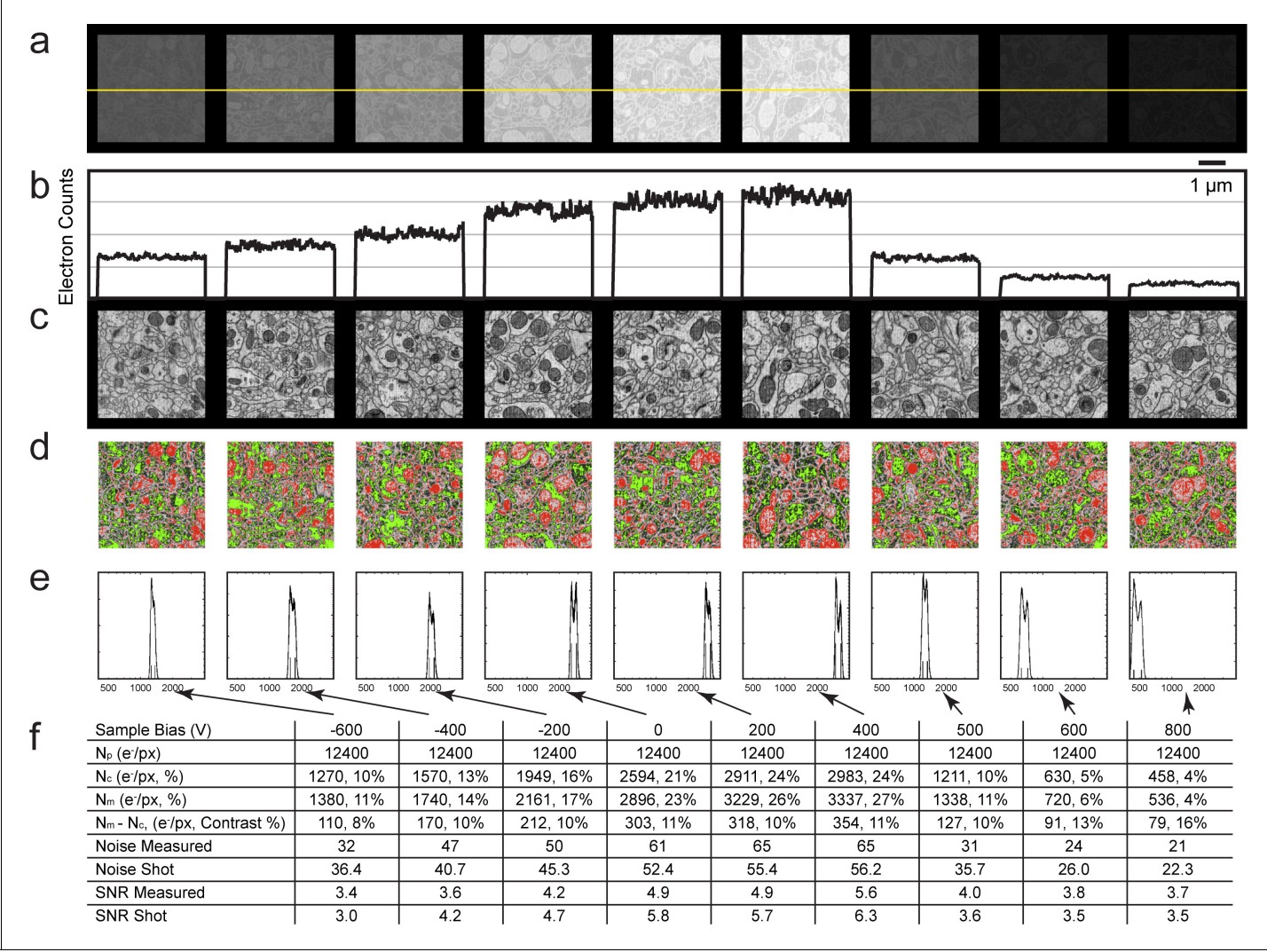

**Figure 12.** Analysis of specimen bias effects in FIB-SEM applications. (**a**) Series of constant grey scale FIB-SEM images at various bias conditions from −600 to 800 V with identical electron count grey scales. Landing energy was also fixed at 1.2 keV. (**b**) Intensity profile of the yellow line in (**a**) illustrates the electron counts as a function of bias voltage over a 0 to 4000 count range. (**c**) Images of (**a**) after grey scale adjustment. (**d**) Red and green thresholds indicate $N_m$ and $N_c$. Histogram of counts per pixel (**e**), Sample bias, SNR and contrast (**f**) of the corresponding images. Scale bar, 1 μm in (**a**), (**c**), and (**d**).

The following figure supplement is available for figure 12:

**Figure supplement 1.** Enlarged figures from *Figure 12* to illustrate the manual thresholding method of $N_m$ and $N_c$ estimates.

## Model results on reference compounds

We first apply the model to three well-characterized reference compounds to confirm its validity, before considering the neuropil sample where the membrane-staining percentage of osmium is not accurately known. The reference compounds are pure gold, epoxy (the embedding plastic used for the biological samples), and Tetrakis (triphyenylphosphine) platinum Pt[$(C_6H_5)_3$P]$_4$, a compound with a known 16% density by weight of platinum (atomic number Z = 78), which we expect to scatter electrons similar to osmium-stained lipids (Z = 76). *Figure 13* shows the resulting model energy-angle distribution of backscattered electrons for the three cases, at 1.2 keV primary electron energy. For pure gold, 40% of the incoming electrons are backscattered, a value in good agreement with previous measurements (*Assa'd and El Gomati, 1998*). For epoxy, $C_{21}H_{25}O_5$, the absence of high Z

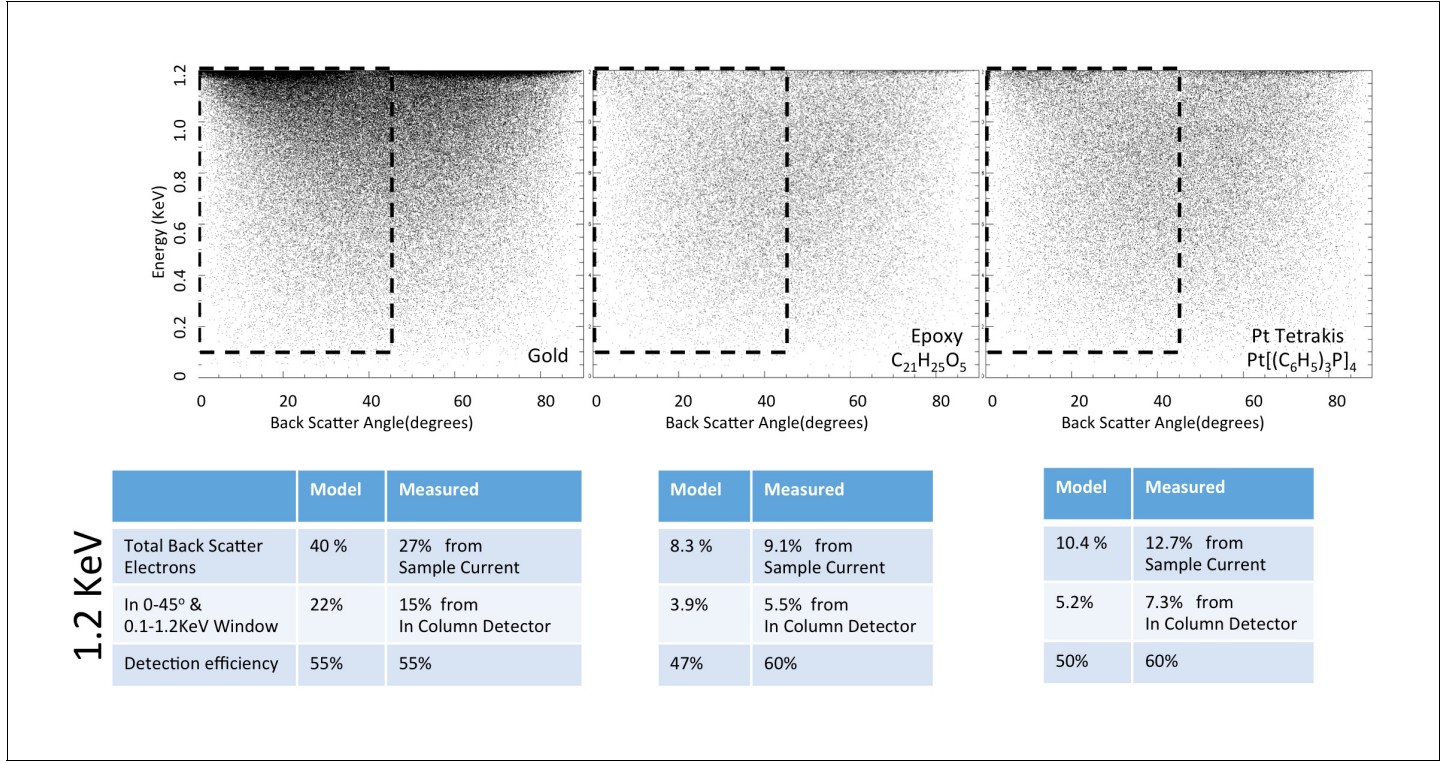

**Figure 13.** Energy and angle distribution of backscattered electrons for 1.2 keV primary electrons impinging on a reference gold, epoxy and Pt Tetrakis sample with 15% by weight of high Z element. The measured and modeled values are in rough agreement (within 30%). Both suggest that about 50% of the backscattered electrons are detected.

atoms reduces backscattering to a base of 8.3% of the incoming beam. The 16% wt loading of Pt in the reference $Pt[(C_6H_5)_3P]_4$ compound boosts the backscattered fraction to 10.4%.

## Detection efficiency model

Not all the back-scatted electrons will be detected by the InLens detector. If such a detector could detect all electrons up to 45 degrees from the axis, then the distribution could be integrated to get a corresponding ~50% detection efficiency. This estimated 50% collection efficiency is more accurately described by electrostatic ray tracing of backscattered electrons of different energies and angles. Our modeling results (*Scientific Instrument Services, 2011*) are shown graphically in *Figure 14*. For the geometry of 3 mm working distance, the electrostatic potential near the end of the objective lens and +600 V sample bias, the model shows the energy/angle values of electrons that enter the objective aperture vs. those that do not. This shows a cut-off that can indeed be approximated by 45° threshold. Also, the +600 V sample bias ensures that the secondary electrons (whose energies are below 50 eV) are excluded from detection and do not contribute artifacts to the image. However, further increase in bias beyond +600 V would reduce signal intensity, as backscattered electrons are pulled away from the detector or even back onto the sample, depending on their energy. This prediction is consistent with experimental results shown in *Figure 12b*.

## Detection efficiency measurement

The ratio of InLens detected electrons to the number of backscattered electrons provides an alternate measure of detection efficiency. The full backscattered current can be computed as the difference between primary beam current and the measured sample current, since the positive bias suppresses any secondary electron current. In this way we measure the backscattered current from the reference gold sample, *Figure 15*, at ~27% of the primary beam current, varying by ±3–4% depending on the orientation of the gold crystal domains. Our experimentally determined ratio of

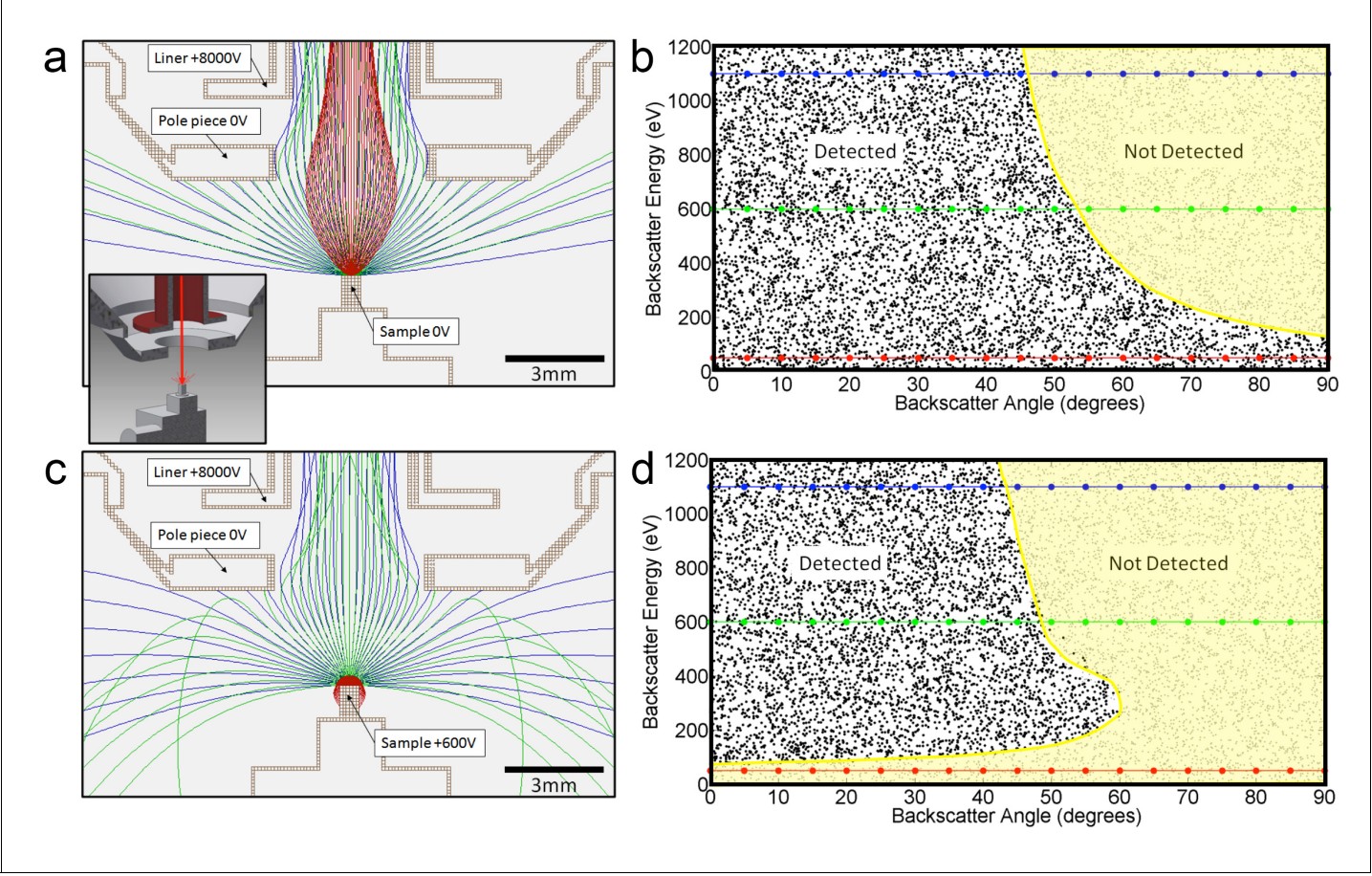

**Figure 14.** Positive bias applied to the sample suppresses secondary electrons but leaves most backscattered electrons unaffected. In effect, this converts the SEM's InLens 'secondary' detector into an efficient high-bandwidth backscatter detector well suited for FIB-SEM. (**a**) Electron flight simulation (using SIMION software package) showing paths of 1.1 keV (blue), 600 eV (green), and 50 eV (red) electrons emerging from an unbiased sample surface at a range of angles (−90° to 90° in 5° increments). After emerging from the sample surface, electrons are attracted by the +8 kV liner tube of the Zeiss Gemini column. (**Kumagai and Sekiguchi, 2009**) Inset shows the 3D CAD model that this simulation was based on. (Only electrostatic elements were modeled, not the magnetic field of the objective lens.) The lowest energy electrons (red) are all funneled up the column and thus potentially impact the InLens detector of the Gemini column. Here, we assume that all electrons that make it into the liner tube are in fact detected. For higher energy electrons (blue and green) only the central angles (−45° to 45°) are detected. (**b**) 10,000 electrons paths were simulated in the same model, covering a uniform range of starting energies (0 to 1.2 keV) and angles (−90° to 90°), allowing the detected vs. non-detected regions in energy vs. angle space to be plotted. All electrons below 100 eV are detected (i.e. make it into the liner tube) in this 0 V bias condition. 1.1 keV (blue), 600 eV (green), and 50 eV (red) electrons simulated in (**a**) are superimposed on this plot for cross-reference. (**c**) Same simulation as in (**a**) but with +600 V bias on the sample. This has a dramatic effect on the paths of the lowest energy (red) electrons, which are all pulled back onto the sample. (**d**) This filtering of low-energy (secondary) electrons is clearly seen in this corresponding energy vs. angle detection plot. Comparing (**d**) to (**b**) one can see that sample bias has only slight effect on the detection of higher energy electrons. The overall detection efficiency is determined by this acceptance mask, combined with the backscattered electrons distribution of *Figure 13*.

backscattered current to primary current is somewhat lower than the 40% predicted by the model and suggested by previous measurements (*Assa'd and El Gomati, 1998*). Nevertheless, it gives confidence that we can roughly estimate the magnitude of the electron signals using this technique. We speculate that the residual discrepancy could be explained by additional tertiary electron currents generated by the backscattered electrons that impinged on the pole tip and are reabsorbed by the positively-biased sample. A fraction of the backscattered electrons enter the InLens detector. The corresponding measured InLens current is about half of the total backscatter current (55% detector efficiency at 1.2 keV, *Figure 15*), consistent with the model of masking away the larger angled backscattered electrons.

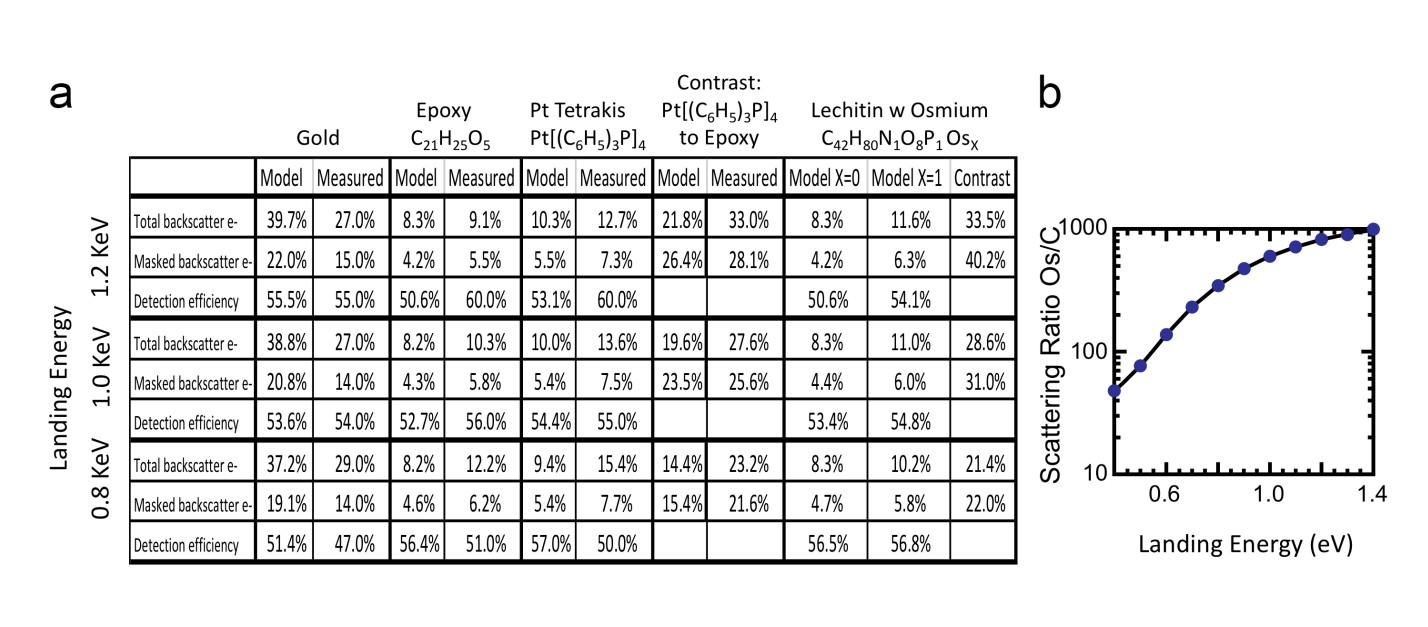

| | | Gold | | Epoxy $C_{21}H_{25}O_5$ | | Pt Tetrakis $Pt[(C_6H_5)_3P]_4$ | | Contrast: $Pt[(C_6H_5)_3P]_4$ to Epoxy | | Lechitin w Osmium $C_{42}H_{80}N_1O_8P_1Os_X$ | | |
|---|---|---|---|---|---|---|---|---|---|---|---|---|
| | | Model | Measured | Model | Measured | Model | Measured | Model | Measured | Model X=0 | Model X=1 | Contrast |
| **1.2 KeV** | Total backscatter e- | 39.7% | 27.0% | 8.3% | 9.1% | 10.3% | 12.7% | 21.8% | 33.0% | 8.3% | 11.6% | 33.5% |
| | Masked backscatter e- | 22.0% | 15.0% | 4.2% | 5.5% | 5.5% | 7.3% | 26.4% | 28.1% | 4.2% | 6.3% | 40.2% |
| | Detection efficiency | 55.5% | 55.0% | 50.6% | 60.0% | 53.1% | 60.0% | | | 50.6% | 54.1% | |
| **1.0 KeV** | Total backscatter e- | 38.8% | 27.0% | 8.2% | 10.3% | 10.0% | 13.6% | 19.6% | 27.6% | 8.3% | 11.0% | 28.6% |
| | Masked backscatter e- | 20.8% | 14.0% | 4.3% | 5.8% | 5.4% | 7.5% | 23.5% | 25.6% | 4.4% | 6.0% | 31.0% |
| | Detection efficiency | 53.6% | 54.0% | 52.7% | 56.0% | 54.4% | 55.0% | | | 53.4% | 54.8% | |
| **0.8 KeV** | Total backscatter e- | 37.2% | 29.0% | 8.2% | 12.2% | 9.4% | 15.4% | 14.4% | 23.2% | 8.3% | 10.2% | 21.4% |
| | Masked backscatter e- | 19.1% | 14.0% | 4.6% | 6.2% | 5.4% | 7.7% | 15.4% | 21.6% | 4.7% | 5.8% | 22.0% |
| | Detection efficiency | 51.4% | 47.0% | 56.4% | 51.0% | 57.0% | 50.0% | | | 56.5% | 56.8% | |

Landing Energy

**Figure 15.** Comparison of Monte Carlo model and measurements: effects of primary landing energy on total and detected-fraction backscattered electrons. (a) Table of results of Monte Carlo model and measurements of backscattered electrons at different landing energies, for the three reference samples and a model for an osmium-stained lipid. At lower landing energies, the contrast between stained and unstained sample drops rapidly, consistent with the increased carbon cross-section vs that of osmium at low energies. Curve is from NIST elastic scattering data. (b). With typical sample stoichiometry of 100 carbon atoms per high Z stain atom, the signal vanishes rapidly below 600 eV.

The backscattered electron signal was also determined for epoxy and for $Pt[(C_6H_5)_3P]_4$, as shown in **Figure 16a**. The InLens current and total sample current values of these two samples are tabulated in **Figure 15**; all experimental values are consistent with a 50–60% InLens backscatter detection efficiency and the modeled signal strength.

## Calibration of sample osmium concentration

A quantitative comparison can be extended from these reference samples to neuropil in **Figure 16b**. This osmium-stained cell membrane parts of the sample have about the same signal and contrast as Pt Tetrakis in epoxy, suggesting that the sample has approximately the same 16% by weight heavy metal fraction. For a representative estimate of stained plasma membrane signal, we assume a composition close to one osmium atom for each molecule of lipid ($X_{Os}$ = 1, stained) (**Riemersma, 1968**; **Sousa et al., 2008**). Using lecithin, a common membrane lipid with a composition of $C_{42}H_{80}N_1O_8P_1Os_{1.0}$ as a representative lipid would predict 20% by weight of the osmium stain. The measured signal is almost identical to the $Pt[(C_6H_5)_3P]_4$ with its 16% Pt by weight. This is consistent with $Os_{0.8}$ ($X_{Os}$ = 0.8), a reasonable value given the complexities that determine staining concentration of heavy metal stain in a real cell membrane.

## Effect of landing energy on signal and contrast

The dependence of the fraction of backscattered electrons on the energy of the primary beam is summarized in the table of **Figure 15a** for both modeled and measured values, for the three reference samples and for a modeled lipid-osmium sample. The parameter of interest is the staining contrast for a known percentage of highly scattering heavy metal. This is tabulated for contrast between epoxy to $Pt[(C_6H_5)_3P]_4$ for both modeled and measured values, and also between lipid membrane with 0 and 1 molarity of osmium stain. In all cases, the best contrast is at the highest energies of 1.2 keV, and drops over a factor of 2 with lower landing energies of about 600 eV, reflecting the increased relative backscattering cross-section of carbon vs. osmium at low energies, **Figure 15b**. For typical samples with ~100 low Z (mainly hydrogen, carbon and oxygen) atoms for every high Z

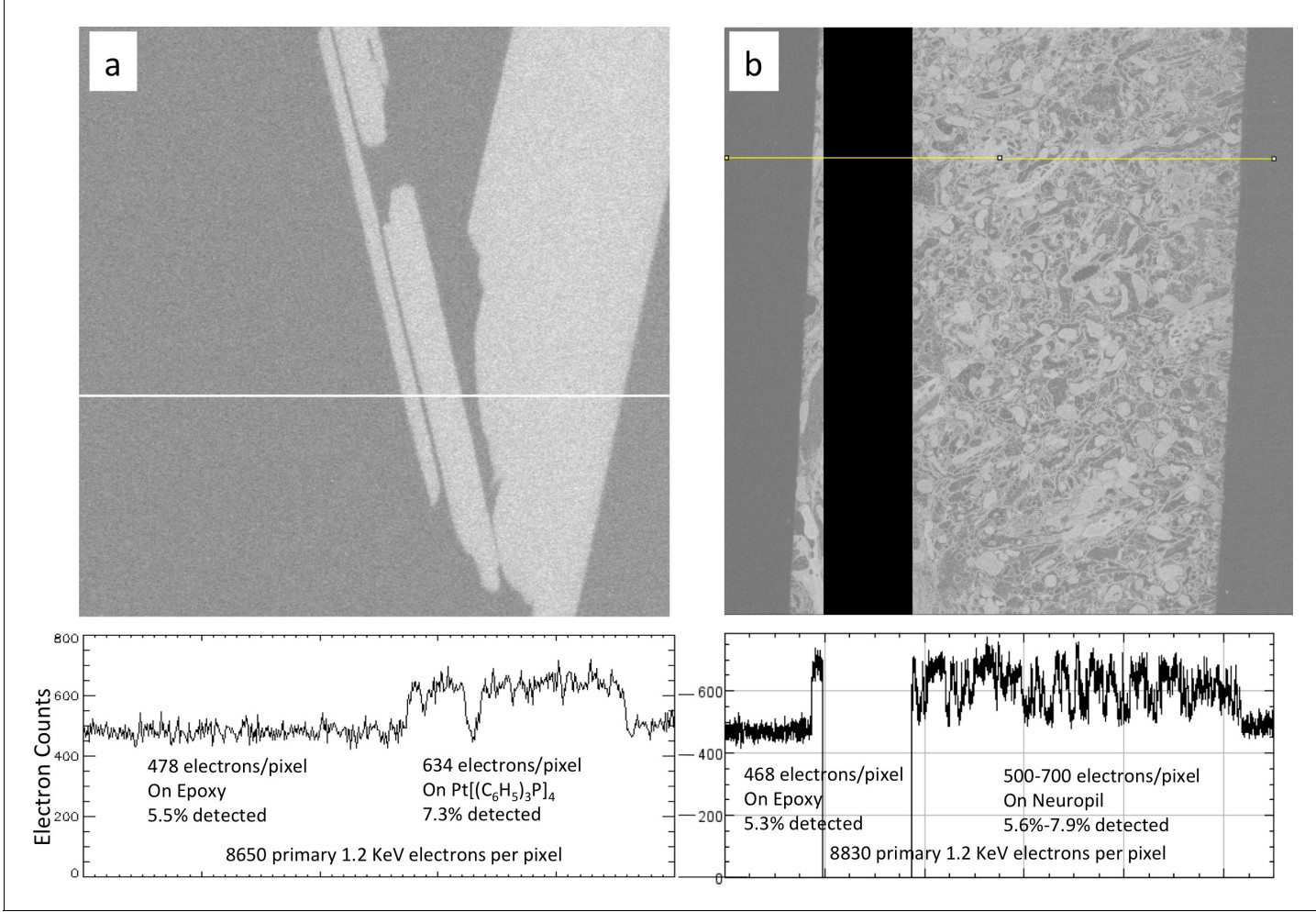

**Figure 16.** Calibration of sample osmium concentration using reference Pt Tetrakis standard. (a) Signal from epoxy and an embedded Pt Tetrakis sample which yield about 5.5% and 7.3% detected backscatter electrons, respectively, from the 8650 incoming electrons on each pixel. The average differential of ~150 electrons is subject to shot noise of about 22 electrons, giving an SNR of 8. One can see in (b) that typical fly neuropil prepared using the PLT procedure with a non-quantified osmium stain concentration has about the same contrast and SNR and can be calibrated against this Pt Tetrakis standard.

osmium atom, this explains why contrast vanishes below 600 eV, when the weighted contrast ratio falls to unity.

## Effect of landing energy on the point spread function

The landing energy determines the resolution, since higher energy electrons scatter further and explore larger volumes of the sample before they backscatter. This can be quantified by the Monte Carlo model in *Figure 17a,b*, which shows such a distribution of scattering locations for primary electron energies of 0.8 and 1.2 keV in a sample not stained with osmium, and approximates the Point Spread Function (PSF) for signal. The spatial distribution difference between stained and unstained samples is relatively small, *Figure 17a,c*. These figures of scattering location can be mis-leading in estimating the contribution to PSF arising from the distribution of scattering locations. Any differential contrast will also be heavily weighted by the higher energy scattering events, which are concentrated even more to the center in the red (highest energy backscattered electrons) region.

A more direct way to model the lateral or vertical resolution is to use a step edge (in x and z) in concentration in the sample model, and move it across the beam to get a contrast profile. Four such

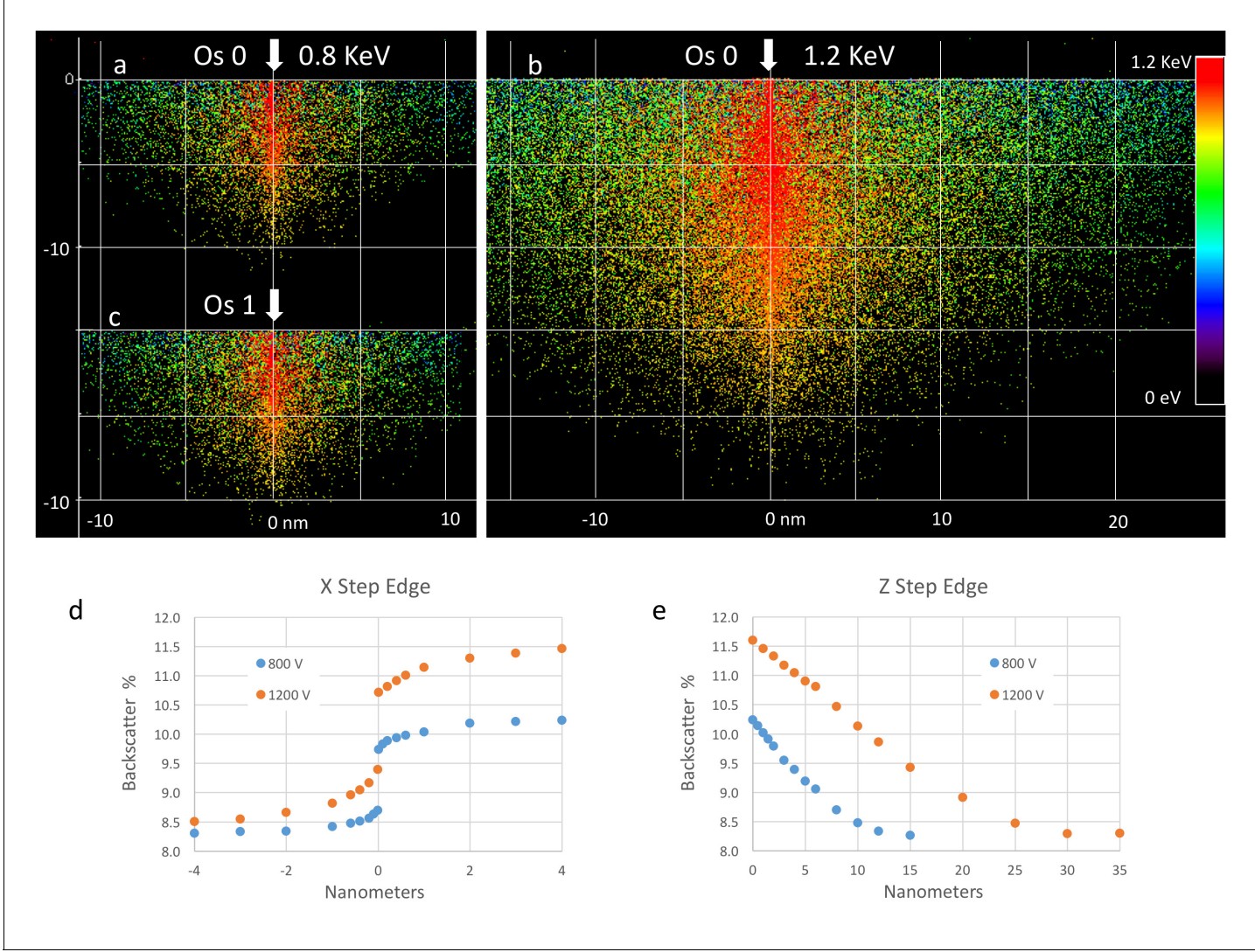

**Figure 17.** Effect of landing energy on the point spread function based on Monte Carlo simulations. (**a,b,c**).Monte Carlo simulation of backscattered electrons at different landing energies. Lower landing energy generates smaller sampling volume (implying better resolution), but with reduced contrast and SNR, as discussed in (**d,e**). A better measure of resolution is to model a step edge in staining from $x_{Os} = 0$ to $x_{Os} = 1$. The step can be lateral and shows a small transition region of less than 1 nm for about half of the signal. The sensitivity to a depth transition in staining is more gradual, with a P50 value of 5 nm for 800 eV landing energy and 11 nm for 1.2 keV landing energy. Note that the simulations assume a primary beam with zero lateral spread. The actual resolution is convolved by the actual beam width.

curves are shown also for the exemplary 800 eV and 1200 eV cases in *Figure 17d,e*. About half of the step in contrast when moving laterally from osmium-free to osmium-stained region takes place within 1.0 nm. This contribution is a result of backscattering that occurs with little or no prior lateral scattering. Such laterally narrow PSF with a larger depth dependent on landing energy was also suggested previously (*Hennig and Denk, 2007*). From *Figure 17e*, we note that a 50% change in contrast occurs when the boundary between the osmium-rich and osmium-poor layers is 5 nm deep for 800 eV electrons, and 11 nm for 1.2 keV electrons. In choosing operating conditions, consideration should be given to the tradeoff between the best achievable resolution at a given beam current, and the higher current demanded by throughput needs. In optimizing the data acquisition, it is also important to consider a range of landing energies, which affects the relative strength of the heavy metal stain signal and the spatial point spread function of the probing electron beam.

## Signal and resolution summary

By comparing these simulations and the reference material measurements with the actual data (*Figure 12*), we confirm that the signals with 50% collection efficiency of the detector are all consistent. For example, the 7.3% and 5.5% backscatter values in *Figures 15a* and *16* of the 1.2 keV landing energy data for Pt[(C$_6$H$_5$)$_3$P]$_4$ is roughly consistent with the 5–6% backscattered fraction seen in the +600 V biased samples of *Figure 12*. The differential contrast between the reference osmium-stained and non-osmium-stained lecithin backscatter of 40% in the $x_{Os}$ = 1 model is higher than the observed 13%, but this can be explained by a reduced osmium concentration of 6.5% by weight ($X_{Os}$ = 0.3) over the beam-sampled volume. Overall, we conclude that the signals are reasonably well understood, and optimized according to expected signal.

## Sample preparation

Conventional biological sample preparations optimized for serial sectioning can be directly applied to 3D FIB-SEM with minor modifications. Both chemical fixation with mixed aldehydes and high-pressure freezing followed by freeze substitution yielded successful results on FIB-SEM.

### *Drosophila* brain

Two different methods were used to prepare Drosophila brain tissue imaged by FIB-SEM. For one approach, the head of a 5-day-old adult female CantonS G1xw1118 Drosophila was cut into 200 µm slices with a Leica VT1000 microtome in 2.5% glutaraldehyde and 2.5% paraformaldehyde, in 0.1 M cacodylate at pH 7.3. The vibratome slice was fixed for a total of 10–15 min, transferred to 25% aqueous bovine serum albumin for a few minutes, and then loaded into a 220 µm deep specimen carrier and high-pressure frozen in a Wohlwend HPF Compact 01 High-Pressure Freezing Machine (Wohlwend Gmbh). The brain was then freeze-substituted in a Leica EM AFS2 system in 1% osmium tetroxide, 0.2% uranyl acetate and 5% water in acetone with 1% methanol, for three more days (*Takemura et al., 2015*). At the end of freeze-substitution, the temperature was raised to 22°C and tissues was rinsed in pure acetone, then infiltrated, and embedded in Durcupan epoxy resin (Fluka).

Alternatively, whole *Drosophila* brains were fixed in 2.5% formaldehyde and 2.5% glutaraldehyde in 0.1 M phosphate buffer at pH 7.4 for 2 hr at 22°C. After washing, the tissues were post-fixed in 0.5% osmium tetroxide in ddH$_2$O for 30 min at 4°C. After washing and *en bloc* staining with 0.5% aqueous uranyl acetate for 30 min, a Progressive Lowering Temperature (PLT) procedure started from 1°C when the tissues were transferred into 10% acetone. The temperature was progressively decreased to −25°C, while the acetone concentration was gradually increased to 97%. The tissue was fixed in 1% osmium tetroxide and 0.2% uranyl acetate in acetone for 32 hr at −25°C. After PLT and low-temperature incubation, the temperature was increased to 22°C, and tissues were rinsed in pure acetone, then infiltrated, and embedded in Poly/Bed 812 (Luft formulation).

### Mammalian brain

All vertebrate procedures were performed strictly in accord with protocols approved by the UNC Animal Use and Care Committee. Briefly, after induction of deep anesthesia with sodium pentobarbital (80 mg/kg IP), adult mice (male C57/BL6J, from Charles River) were intracardially perfused with a mixture of 2% glutaraldehyde/2% depolymerized paraformaldehyde, after a brief saline rinse. Brains were removed and postfixed in the same mixture overnight at 4°C. Blocks of tissue including the striatum were cut at 50 µm on a vibratome and sections were collected in phosphate buffer (0.1 M, pH 7.4). Sections were incubated 30 min in 0.1% CaCl$_2$, then processed for reduced osmium according to the Graham Knott protocol (*Knott et al., 2008*), then treated with 2% samarium trichloride and 1% uranyl acetate in maleate buffer pH 6.0, prior to dehydration and infiltration in Durcupan resin. Sections were sandwiched between two layers of ACLAR plastic between glass slides, and polymerized 48 hr at 60°C.

### *Chlamydomonas reinhardtii* cell preparation

*Chlamydomonas reinhardtii* cells, 4A+ strain (*mt*+) in the 137c genetic background obtained from Jean-David Rochaix (University of Geneva), were grown heterotrophically on TRIS-acetate-phosphate (TAP) medium in the dark at room temperature. Whole cells were sedimented, lightly fixed with

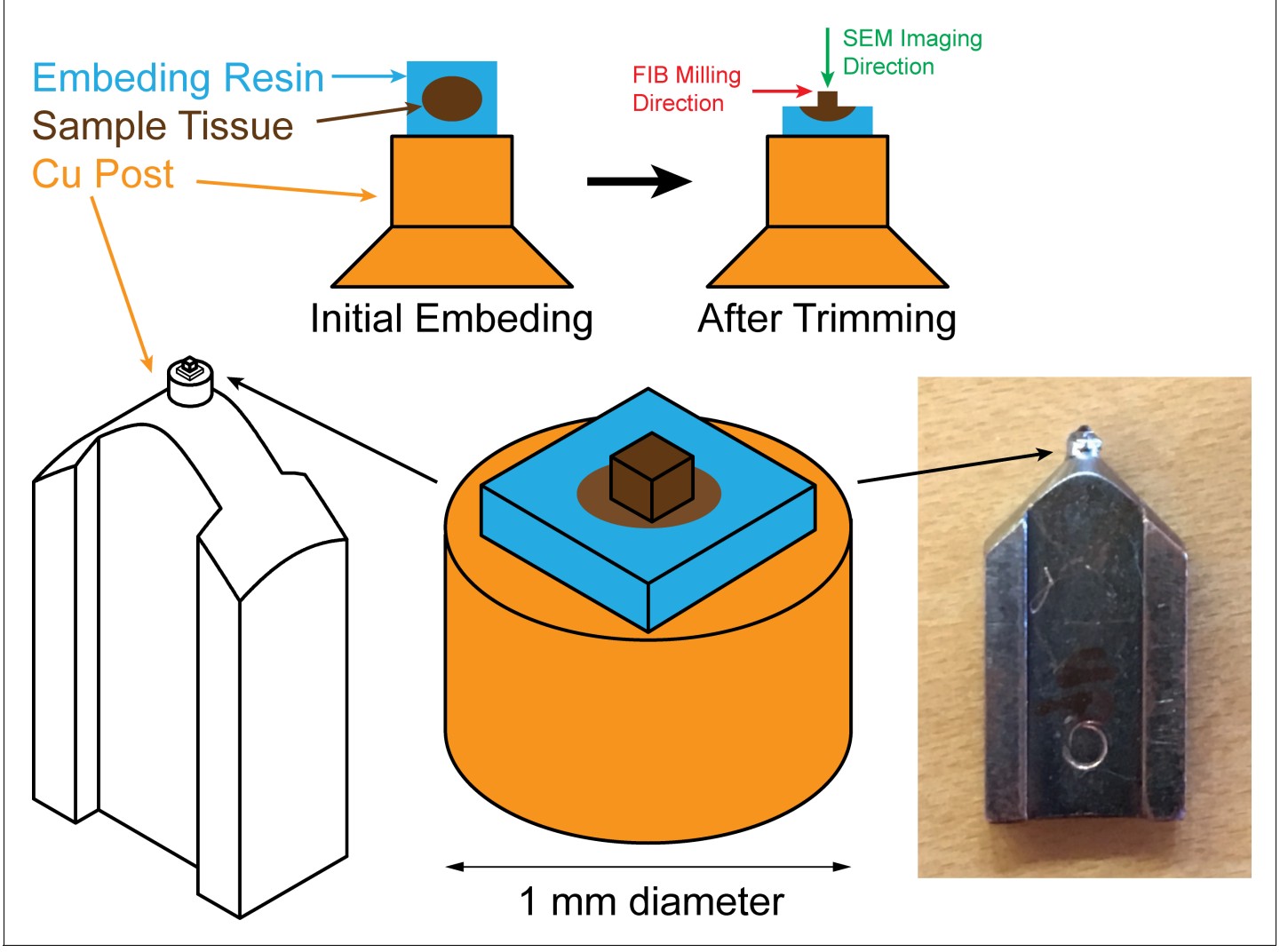

**Figure 18.** Sample mounting and trimming diagrams for FIB-SEM: sample (colored in brown) embedded in resin (colored in blue) is mounted onto a Cu stud (colored in orange), embedding resin is then trimmed off to expose the sample. The green arrow pointing down indicates the scanning SEM beam which is perpendicular to the FIB milling direction indicated by the red arrow.

glutaraldehyde (1%) and post-fixed with osmium tetroxide, potassium ferrocyanide, and uranyl acetate prior to dehydration and embedding in hard Durcupan resin.

## Embedding resin considerations for FIB-SEM

Radiation from SEM changes the property of the polymer resins commonly used for embedding biological materials. Post-radiated resins are subject to milling artifacts, including 'streaks' (static line features parallel to the milling direction, shown in *Figure 11c,f*) and 'waves' (dynamic uneven milling bands perpendicular to and traveling along the milling direction). The radiation responses of various resins are drastically different. For example, acrylic-based resins such as LR White and regular Epon (*Figure 11c*) tend to generate severe streaks and waves, whereas Durcupan (*Figure 11d*) can sustain much higher electron beam dose without noticeable artifacts. Various other measures (e.g. protective hard coating as milling cap layer, smaller milling current, etc.) have been developed to mitigate these side effects of electron beam radiation. If an alternate embedding plastic such as Epon is needed, one can mitigate some of the milling artifacts by coating the front face (facing the ion beam) with a 5–10 μm layer of Durcupan. Nevertheless, there is a milling depth limit in order to maintain a uniform z-step thickness. We found that for a z-step of less than 10 nm, the FIB column

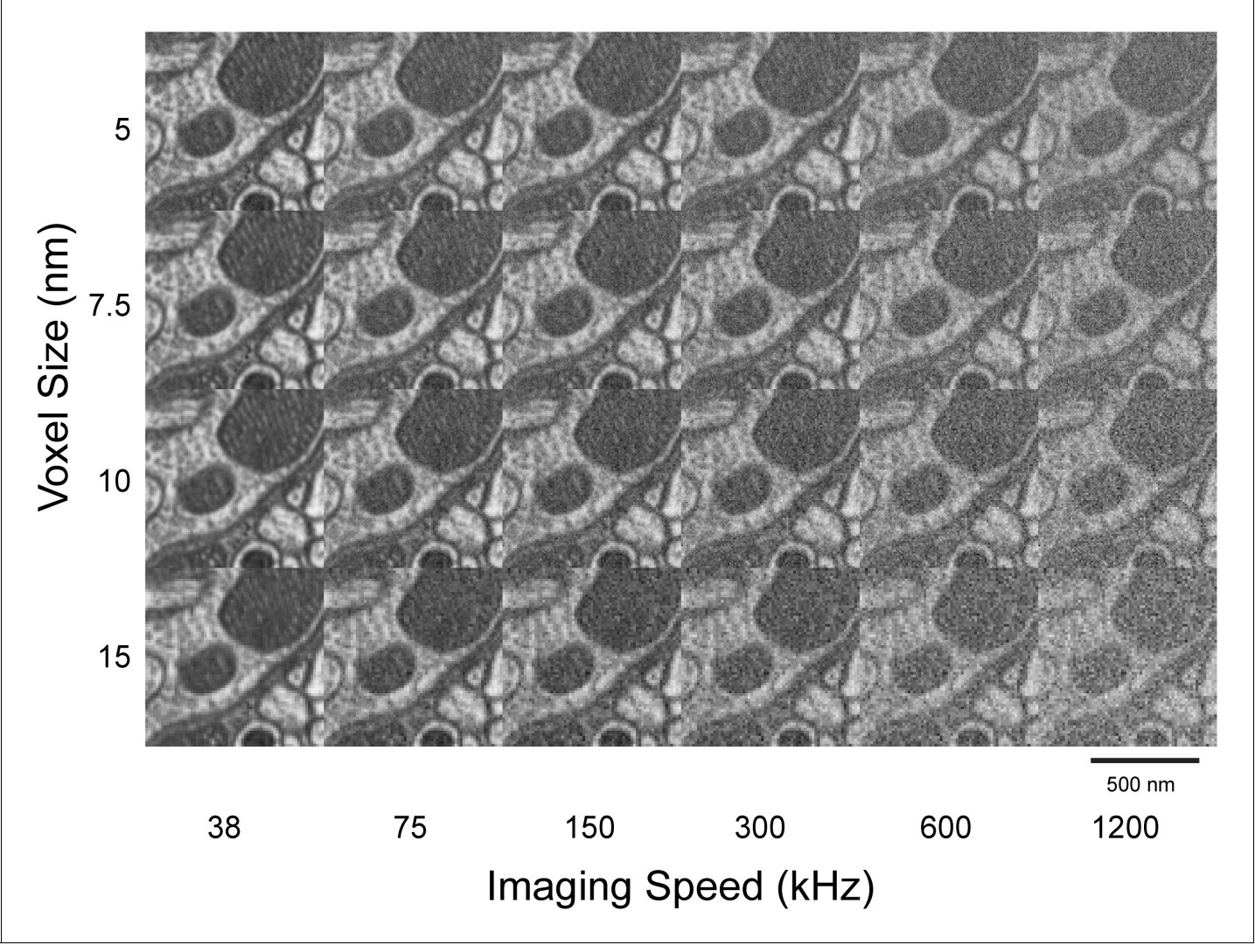

**Figure 19.** Imaging speed and voxel size study to determine optimal condition for neuronal circuit reconstruction. A high-resolution (5 nm) image stack was first acquired at low imaging speed (38 kHz). Corresponding larger voxel and more rapidly acquired images were emulated by binning and adding shot noise through software. A condition of around 8 nm and 300 kHz was found to optimize traceability and throughput. Scale bar, 500 nm.

has difficulty milling more than 100 µm in the beam direction on Durcupan resin (assuming a nominal SEM dose of 30 electrons per $nm^3$). To permit large volume imaging using FIB-SEM, an ultrathick partitioning method has been developed, which not only removes this potential hard barrier for large volume acquisition, but also enables high-throughput parallel 3D imaging with FIB-SEM (*Hayworth et al., 2015*). These 20-µm-thick slabs also require a Durcupan coating on the front side.

## Final trimming and preparation for FIB-SEM

Samples are mounted to the top of a 1 mm metal post, if possible with the metal in contact with the metal-stained sample for better charge dissipation. A small vertical sample post is trimmed to a width of <300 µm and a depth of <200 µm in the direction of the ion beam (*Figure 18*). The trimming is usually guided by optical inspection under a microtome and X-ray tomography data obtained by a Zeiss Versa XRM-510. The limited block-face size ensures a complete removal of the block-face material, rather than forming a trench in the sample. This arrangement improves milling stability by eliminating sidewall effect and back sputtering. A thin layer of conductive material of 10- to 20-nm gold followed by 50- to 100-nm carbon is coated on the trimmed sample using a Gatan

681 High-Resolution Ion Beam Coater. The coating parameters are 6 keV, 200 nA on both argon gas plasma sources, 10 rpm sample rotation with 45-degree tilt.

## FIB-SEM operation conditions

Images were acquired on a Zeiss NVision40 system using 1.1 keV electron beam energy with a sample bias voltage of 0.4 kV, which resulted in a landing energy of 1.5 keV. The probe current was set at ~3 nA and working distance at ~4.5 mm with imaging speed of 1.25 MHz. Images acquired on a Zeiss Merlin system (hybrid with 90 degree mounted FEI Magnum FIB) used slightly different conditions: landing energy was 1.2 keV with 0.6 keV electron beam energy with a sample bias of 0.6 kV. The probe current was set at ~4 nA and working distance at ~3 mm with imaging speed of 4 MHz. The x-y pixel size was 8 nm for both systems unless noted. SEM images were acquired for every 2 nm of material removal. After the final image series were registered using IMOD (*Kremer et al., 1996*) or SIFT plug-in of Fiji (*Schindelin et al., 2012*), every four consecutive images were binned down to one, forming an image stack with isotropic voxels of $8 \times 8 \times 8$ nm$^3$.

The gallium FIB column was operated at 30 keV. A 7-nA probe current was selected for milling with the FEI Magnum column, while a 13- or 27-nA probe current was used in a Zeiss NVision40. A repeated line scan of 0.1 µm pixel and 1-MHz frequency was applied. A 10 Hz PID closed-loop algorithm written in LabVIEW (National Instruments) controlled the line scan position relative to the specimen block-face. To minimize overhead, the milling time was typically set to be less than 20% of SEM imaging time. With a $200 \times 200$ µm$^2$ block-face specimen, the milling time was around 10 s or less for every frame (2 nm z-step) using a 27-nA FIB probe current.

## FIB-SEM control system customization

A customized hardware, control, and software package was developed to enable long-term acquisition on a FIB-SEM system. Major hardware components included a National Instrument signal generation and acquisition system, temperature sensors, a high-voltage isolation current amplifier, and a home-built computer with RAID6 storage. A National Instrument PXIe-1082 chassis, equipped with two PXI-5421, one PXIe-5122, one PXIe-6124, and one PXIe-6259, was connected to the RAID6 computer through a PXIe-PCIe8371 bridge card. Collectively, they provided scan signals for SEM imaging and FIB imaging/milling, as well as SEM image collection and storage. This system was able to acquire data from two SEM channels up to 32k x 32k pixel at 100 MHz. It also recorded machine vital signs, such as ambient temperature, specimen current, and Faraday cup currents, at 10 Hz. Software written in LabVIEW was used to control the entire FIB-SEM operation.

## Optimal voxel size and imaging speed

Voxel size is reciprocally related to imaging speed; the best compromise must be determined for each study. Smaller voxel size and lower imaging speed generate higher quality data sets, but can be impractically slow, depending on the specimen and the biologically relevant sample sizes. The optimal conditions also depend on the specimen preparation, especially the staining conditions and the resulting electron contrast. We performed experiments to determine the optimal-throughput balance between imaging and subsequent analysis for complete circuit reconstruction of *Drosophila* brain. First, we collected a high-quality image stack of *Drosophila* medulla at $5 \times 5 \times 5$ nm$^3$ voxel size, with small current and slow scanning speed. Software binning and shot noise were then added to simulate datasets collected with larger voxels and at faster scanning speed (*Figure 19*). Based on the automated segmentation and human proofreading speed and error rate, we determined that a voxel sampling of 8 nm was optimal for our *Drosophila* connectomics studies. This is at least 50% smaller then Nyquist sampling distance for beam blur associated with currents of up to 4 nA characterized by the edge signal intensity drop-off of gold nanoparticles on a carbon surface (Electron Microscopy Sciences, P.O. Box 550, 1560 Industry Road, Hatfield, PA 19440, USA, Part #79511–01). Dwell times of 0.8 to 3 µs with 12,000 to 48,000 primary beam electrons per pixel gave a useable SNR of 5 or better.

## Higher resolution imaging conditions and constraints

Continuous long-term operation opens up opportunities for studies that require higher resolution. For low-voltage (~1 keV) SEM's operating with 0.1–10.0 nA on the primary beam, the resolution is a

function of current (*Reimer, 1993*). Spherical and chromatic aberration are prime contributors to beam blur, so smaller numerical apertures with smaller beam currents typically improve resolution. Unfortunately, the downside of better resolution is a longer image acquisition time; conversely, a reduced volume size will be accessible in a fixed time. As an example, we have 2.7 nm resolution (measured as 50% rise distance at a step edge) at 0.08 nA vs 5.5 nm at 4.0 nA. To keep the same sampling-to-resolution ratio, there will be a 2x shorter sampling distance or 8x more voxels for 2x higher resolution in a given volume. Furthermore, since the electron dose per voxel needs to be the same for constant signal-to-noise (assuming mainly shot noise), we need to integrate the current on each smaller voxel for ~50x longer. Together this means that doubling the resolution requires a 400x slower volume acquisition rate! Thus, instead of $(100 \ \mu m)^3$, only a $(14 \ \mu m)^3$ cube can be acquired in 100 days. In general for constant signal-to-noise at a given beam current $I(\delta)$ and with voxels scaling with resolution $\delta$, the volume rate is proportional to $dV/dt = \delta^3 * I(\delta)$.

One should choose the trade-off between resolution and volume that is optimal for any given sample and line of enquiry. *Figure 1* illustrates the relationship between imaging resolution and achievable volume with contours of required acquisition time. The vertical axis assumes isotropic x, y, and z resolution. Besides primary beam blur, the resolution must include the vertical and lateral extent of the exploration range of the back-scattered electron. The Monte Carlo trajectory simulation discussed previously shows that the landing energy of the primary electron must also be adjusted to lower energies to reduce the spatial spread to values consistent with the primary electron beam blur. This means landing energies of 600–1000 eV. As mentioned in the Monte Carlo discussion, contrast to the heavy metal stain is lost rapidly below 600 eV, setting a lower limit to the practical operating point. To maintain SNR at the lower contrast will require further reduction of imaging speed.

## Comparison with other EM approaches to high-resolution 3D imaging

How does the high-resolution/large volume approach to FIB-SEM imaging presented here compare with other forms of electron microscopy? It turns out to have a complementary application space. Electron tomography based on high-voltage transmission EM is a standard method used to obtain high-resolution 3D images, typically superior to that of lower energy FIB-SEM imaging. This methodology typically uses ~200–500 nm thick sections. Imaging of thicker volumes requires difficult and time-consuming manual registration of multiple sections. To image thicknesses requiring more than a few sections, FIB-SEM is usually a more practical alternative. If one can sacrifice resolution in one dimension and anisotropic voxels are useful, then either traditional serial section TEM or diamond knife-trimmed block-face SEM might be a more practical alternative. The z resolution of mechanical slicing with a diamond knife can be improved with de-convolution in which the sample is imaged using various landing energies. However, this virtual slicing approach adds a substantial burden in operation and reduces throughput and has not yet been demonstrated for large volume applications. In summary, the technological developments presented here enable FIB-SEM to probe a new domain of considerable biological significance.

## Acknowledgements

We would like to thank the FLYEM team (Steven Plaza, Pat Rivlin, Lou Scheffer, Shinya Takemura, Gary Huang, Toufiq Parag, Juan Nunez-Iglesias, and Ting Zhao) for processing the *Drosophila* connectome and providing examples of the speed and completeness of reconstruction that FIB-SEM can enable. We also thank Richard D Fetter for preparation and provision of various SEM samples, as wells as Kent McDonald for consultation on *C. reinhardtii* sample preparation. We thank David Peale and Patrick Lee for consulting support in designing the system modifications. Analysis of *C. reinhardtii* cells was supported by the U.S. Department of Energy, Office of Science, Basic Energy Sciences, Chemical Sciences, Geosciences, and Biosciences Division under field work proposal SISGRKN. RJW thanks NIH for support by R01 NS-039444. KKN is an investigator of the Howard Hughes Medical Institute and the Gordon and Betty Moore Foundation (through Grant GBMF3070).

## Additional information

### Funding

| Funder | Grant reference number | Author |
| --- | --- | --- |
| Howard Hughes Medical Institute | | C Shan Xu<br>Kenneth J Hayworth<br>Zhiyuan Lu<br>Krishna K Niyogi<br>Eva Nogales<br>Harald F Hess |
| U.S. Department of Energy | Office of Science, Basic Energy Sciences, Chemical Sciences, Geosciences, and Biosciences Division - SISGRKN | Krishna K Niyogi<br>Eva Nogales |
| Gordon and Betty Moore Foundation | GBMF3070 | Krishna K Niyogi |
| National Institutes of Health | R01 NS-039444 | Richard J Weinberg |

The funders had no role in study design, data collection and interpretation, or the decision to submit the work for publication.

### Author contributions

CSX, Conceptualization, Data curation, Software, Formal analysis, Validation, Investigation, Visualization, Methodology, Writing—original draft, Writing—review and editing, conceived, designed, and constructed the custom FIB-SEM systems, designed and implemented the custom FIB-SEM control hardware and software; KJH, Conceptualization, Data curation, Formal analysis, Validation, Investigation, Visualization, Methodology, Writing—original draft, Writing—review and editing, conceived, designed, and constructed the custom FIB-SEM systems; ZL, Writing—review and editing, prepared Drosophila samples; PG, AMH, JGG-C, KKN, EN, Writing—review and editing, prepared C. reinhardtii samples; RJW, Writing—original draft, Writing—review and editing, prepared mammalian brain samples; HFH, Conceptualization, Formal analysis, Supervision, Investigation, Methodology, Writing—original draft, Project administration, Writing—review and editing, conceived, designed, and constructed the custom FIB-SEM systems

### Author ORCIDs

C Shan Xu, http://orcid.org/0000-0002-8564-7836
Eva Nogales, http://orcid.org/0000-0001-9816-3681

### Ethics

Animal experimentation: All of the vertebrate animals were handled according to approved institutional animal care and use committee (IACUC) protocols (#13-258.0) of UNC. UNC's PHS Assurance number is D16-00256 (A3410-01); the AALAC Unit number is 000329.

## Additional files

### Supplementary files

• Source code 1. 3D FIB-SEM LabVIEW codes.

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
