## [Decision Letter]

Thank you for submitting your article "Enhanced FIB-SEM Systems for Large-Volume 3D Imaging" for consideration by *eLife*. Your article has been reviewed by three peer reviewers, and the evaluation has been overseen by a Reviewing Editor and Eve Marder as the Senior Editor. The following individual involved in review of your submission has agreed to reveal his identity: Stephen J Smith (Reviewer #2).

The reviewers have discussed the reviews with one another and the Reviewing Editor has drafted this decision to help you prepare a revised submission.

Three experts reviewed your manuscript, and their assessments, form the basis of this letter. As you will see, all of the reviewers were impressed with the importance and novelty of your work. I am including the three reviews at the end of this letter, as there are many thoughtful, specific, and useful suggestions in them that we think will improve the presentation of your work. In particular, although reviewer #1 suggests changing the order of presentation, the other reviewers did not necessarily agree with this point, so this remains in his/her review for you to consider, but not as a requirement. We appreciate that the level of engineering detail required to fully describe the FIB-SEM system will not fit into the body of a conventional article. That said, we encourage you to take advantage of the generous availability of space for supplemental information and figure supplements that is part of the *eLife* ethos.

We also appreciate that some of the suggestions are substantive and others stylistic. You should, of course, feel free to use your judgment regarding the extent to which you would like to make changes in response to the stylistic suggestions.

Reviewer #1:

The manuscript "Enhanced FIB-SEM Systems for Large-Volume 3D Imaging" by Xu et al. describes an optimization of the focused Ion Beam Scanning EM (FIB-SEM) setup for high-resolution, high-throughput 3D-imaging with a special focus on the application in connectomics. The scope and goal of this work is of great importance to connectomics because high-resolution isotropic data obtained at sufficient speed could ameliorate many of the challenges in data analysis in the field. Furthermore, for certain species only such high-resolution data can provide full wiring diagrams. The authors plausibly document their advances and achievements and I think this is an important and crucial advance in the field.

However, I have a few more substantial concerns with respect to the presentation of the work:

1) I find it rather unfortunate for a methods paper to first have to go through some documentations of methodological success in which without first being introduced to the methodological advances it is not totally obvious where the improvements of the author's method lie. I suggest inverting the description and starting by clearly stating the methodological innovations and their relative significance for throughput, image quality and stability. I would be thrilled to learn about this in an instructive way and then at the end be convinced by successful applications of the methods.

2) The methods are by far not sufficiently detailed. In particular, it would not be possible to reproduce the system as it is described right now. For example – and this is just one of many examples – the computation of the focus index used to do inline image auto optimization (subsection “In-line image auto-optimization”, second paragraph), is only sloppily described, formulae are missing. Similarly, the prompt pausing and seamless restart automated control system is not described to any reproducible detail. It is in my view mandatory for a methods paper to provide all insights and details such that other labs could rebuild and use this system. I think it is also absolutely required to provide all the code used in these systems as a supplementary code base which I could not find in this submission.

3) A clear description of the methodological advances as requested in point 1 is also important because some of the results as presented here are actually not novel; e. g. the authors re-introduce the hot-knife-method which they have published before in Nature Methods. It should be absolutely clear that this is not a new contribution of this paper, even if it of course goes along with expanding the imaging range in the third dimension. Similarly, the closed loop control of the FIB beam is an original contribution by Boergens and Denk – this is cited properly, but still it would be very important to know upfront which innovations are the author's and which are extensions or usage of existing methods. As another example: reading the results one cannot judge the effect of detector optimization on imaging speed. In a methods paper, this should be possible, it's (potentially) a key result.

In summary, again, this is an important development, but as presented I cannot fully judge its advances in a precise and detailed way.

Reviewer #2:

This is an extraordinarily clear, thorough and detailed description of numerous major and very important refinements to the FIB-SEM image acquisition modality. Since FIB-SEM is now widely considered supreme amongst volumetric EM image acquisition methods in important areas of imaging performance, the refinements described here are likely to be highly consequential in several fields of biology. The careful quantitation and modeling of imaging results it presents makes for a rather lengthy presentation, but these lengths are very welcome in view of the novelty of many of the methods introduced, the unfamiliarity of this imaging physics territory to most biologists, and the clear and well-documented absolute superiority of the imaging results this refined methodology brings into reach.

The first part of this manuscript provides fair and reasonable accounts of the strengths and weaknesses of the novel methodology to alternative 3D EM imaging methods. The authors make very compelling arguments that their "enhanced FIB-SEM" methods allow access to regions of a high-resolution vs. large-volume imaging space that are highly significant to current very active areas of neural network neuroscience ("connectomics") and cell / tissue biology. Many of the example images in this methods-focused manuscript indeed reflect the field's first-ever views of previously inaccessible aspects of biological structure. This manuscript thus leaves little doubt that the enhancements introduced and analyzed here will have transformative impact.

One might at first question the need for such detailed description of engineering advances so numerous and imposing as to place the methodology beyond the reach of all "normal" biology laboratories (it is well known that the HHMI Janelia campus enjoys a uniquely high level of engineering support infrastructure). Even allowing, however, that replication and wide adoption of this methodology might be far off in the future, this description in my opinion is likely to be helpful to biologists and technology companies striving to advance the relevant technologies. It is also to be expected that unique reference datasets based on "enhanced FIB-SEM" will soon be published and made available by Janelia investigators, so it will be important for potential "consumers" of these datasets to understand thoroughly the means by which these datasets were acquired. Last but not least, even if "enhanced FIB-SEM" images remain for a time relatively rare, they are likely to serve immediately as "gold standards" against which more accessible volumetric EM methods can be evaluated (as the authors point out).

Reviewer #3:

In this study the authors present new methods for high-resolution imaging with isotropic voxels using the FIB-SEM. This is an important contribution to the field of connectomics, and sharing information about the technique with the community will have high impact. The authors have presented the parameters that are crucial to the success of this imaging modality. They show that with careful monitoring of the ion beam and implementation of a closed loop feedback, it is possible to overcome some of the challenges of this method. They also introduce new methods to quantify signal in terms of electron counts. These advances are tested on a few different model organisms with great results.

Just a few suggestions for improvement of the text:

1) The authors mention multiple "failure modes" during operation of the FIB, for which different strategies were devised. It would be useful to list all these "failure modes" in table, with the appropriate solution and an estimate of the contribution of each of these failure modes during an example run.

2) A method for calculation of SNR is introduced. Here the SNR is proportional to the difference in detected electrons from membrane and cytoplasm. It is however unclear how pixels are classified as membraneous or cytoplasmic.

3) In Section 2 Detection efficiency model, the modeling shows that at a bias of +600V the path of 50eV secondary electrons (red) are shown to be deflected back to the stage. It would be useful to show that beyond the 600V bias, a further increase in bias will in fact start attracting higher-energy back scattered electron away from the detector, as is noticed experimentally in Figure 9.

4) Regarding the paragraph about competing methods in the Introduction, it would be better to slightly change the wording to make clear that these statements are about the current state-of-the-art. (One can't rule out further progress in the future.)

---

## [Author Response]

[…] Reviewer #1:

[…] I have a few more substantial concerns with respect to the presentation of the work:

1) I find it rather unfortunate for a methods paper to first have to go through some documentations of methodological success in which without first being introduced to the methodological advances it is not totally obvious where the improvements of the author's method lie. I suggest inverting the description and starting by clearly stating the methodological innovations and their relative significance for throughput, image quality and stability. I would be thrilled to learn about this in an instructive way and then at the end be convinced by successful applications of the methods.

Reviewer makes an important point, leading us to reconsider the order of presentation. We agree that a traditional "methods" paper should begin with descriptions of methodological innovations followed by results. The problem is that the manuscript is directed toward multiple audiences. The majority of our readership is likely to constitute biologists, who are more interested in the new capabilities of our FIB-SEM system and “the clear and well-documented absolute superiority of the imaging results” (reviewer #2), rather than the nitty-gritty details of the engineering. Accordingly, we felt it best to leave the overall structure as is. Nevertheless, reviewer's point is legitimate. Accordingly, we have added an overview of the technological improvements of this work at the beginning of “Results and Discussion”, to provide readers with a clear picture of the novelty of our work relative to prior arts. Readers who are especially interested in the technical aspects can now find detailed descriptions in a later part of the manuscript and supplemental materials.

2) The methods are by far not sufficiently detailed. In particular, it would not be possible to reproduce the system as it is described right now. For example – and this is just one of many examples – the computation of the focus index used to do inline image auto optimization (subsection “In-line image auto-optimization”, second paragraph), is only sloppily described, formulae are missing. Similarly, the prompt pausing and seamless restart automated control system is not described to any reproducible detail. It is in my view mandatory for a methods paper to provide all insights and details such that other labs could rebuild and use this system. I think it is also absolutely required to provide all the code used in these systems as a supplementary code base which I could not find in this submission.

We agree that some of the methods are not sufficiently detailed for a typical biology lab to replicate the FIB-SEM system presented in the paper. We have revised the “FIB-SEM system customization for continuous long-term acquisition” section to include more detailed descriptions. For example, the formula to calculate the focus index has been added, and description of the software implementation of prompt pausing and seamless restart has been expanded. However, most of the customization is through software control which is not itself innovative, but instead represents a specific implementation of standard engineering techniques. For example, statistical process control methods are common practice in industrial production environments. There are many ways to implement or tailor the control bands, based on end user’s specific need. Therefore, we only pointed out this is one of the many aspects one should include, without going into the exact details of our implementation. Focus index, for another example, is a generic index number to describe how “sharp” an image is. It too can be calculated in many different ways. Binding et al. presented yet another elegant method for this purpose in their work which we have cited. Concordant with HHMI Janelia's mission, our engineering solutions are open, and we encourage interested outsiders to email us or visit our facility to gain independent expertise.

3) A clear description of the methodological advances as requested in point 1 is also important because some of the results as presented here are actually not novel; e. g. the authors re-introduce the hot-knife-method which they have published before in Nature Methods. It should be absolutely clear that this is not a new contribution of this paper, even if it of course goes along with expanding the imaging range in the third dimension. Similarly, the closed loop control of the FIB beam is an original contribution by Boergens and Denk – this is cited properly, but still it would be very important to know upfront which innovations are the author's and which are extensions or usage of existing methods. As another example: reading the results one cannot judge the effect of detector optimization on imaging speed. In a methods paper, this should be possible, it's (potentially) a key result.

We have revised the manuscript to include an overview of the technological improvements of this work at the beginning of the paper. In this section, we distinguish novel developments versus prior arts (such as the hot-knife method and closed-loop control of the FIB beam), which are now explicitly identified and referenced.

In summary, again, this is an important development, but as presented I cannot fully judge its advances in a precise and detailed way.

Reviewer #3:

[…] Just a few suggestions for improvement of the text:

1) The authors mention multiple "failure modes" during operation of the FIB, for which different strategies were devised. It would be useful to list all these "failure modes" in table, with the appropriate solution and an estimate of the contribution of each of these failure modes during an example run.

Figure 9 has been added to outline typical failure modes with different frequencies of occurrence. Corresponding customized solutions were also included.

2) A method for calculation of SNR is introduced. Here the SNR is proportional to the difference in detected electrons from membrane and cytoplasm. It is however unclear how pixels are classified as membraneous or cytoplasmic.

The derivation of detected electrons from membrane and cytoplasm is a somewhat subjective manual operation. We have added Figure 12—figure supplement 1 to explain the details. For example, in order to obtain electron counts of membranes in Figure 12 threshold red level was manually adjusted until about 50% of the cell membranes (outer boundaries) were covered in red. Assuming the membrane electron counts have a particular distribution, such threshold value would represent the median (also the mean if it is Gaussian) of that distribution. Similarly, the cytoplasm electron counts are obtained by a green threshold so that about 50% of the interior cell area (excluding mitochondria) are covered in green, approximately the median of electron counts originating from the cytoplasm. Due to non-uniformity and other issues, this approach is only an estimate. The comparison, though not rigorous, serves to demonstrate the changes in SNR as a function of sample bias voltage. In addition, it serves as an experimental reference for our Monte Carlo simulations.

3) In Section 2 Detection efficiency model, the modeling shows that at a bias of +600V the path of 50eV secondary electrons (red) are shown to be deflected back to the stage. It would be useful to show that beyond the 600V bias, a further increase in bias will in fact start attracting higher-energy back scattered electron away from the detector, as is noticed experimentally in Figure 9.

Reviewer is correct. We have added “However, further increase in bias beyond +600 V would reduce signal intensity as backscattered electrons are pulled away from the detector or even back onto the sample, depending on their energy. This prediction is consistent with experimental results shown in Figure 12” at the end of Section 2 “Detection efficiency model” (old Figure 9 becomes Figure 12 after renumbering).

4) Regarding the paragraph about competing methods in the Introduction, it would be better to slightly change the wording to make clear that these statements are about the current state-of-the-art. (One can't rule out further progress in the future.)

We agree. We have revised to the manuscript to read “In contrast, the current state-of-the-art technologies based on diamond knife sectioning or diamond knife block-face removal, lose consistency when attempting z steps between adjacent images below 20 nm.”